# Cryptic *MYC* insertions in Burkitt lymphoma: New data and a review of the literature

**Renata Woroniecka**[1]*, **Grzegorz Rymkiewicz**[2°], **Lukasz M. Szafron**[3°],
**Katarzyna Blachnio**[2], **Laura A. Szafron**[3], **Zbigniew Bystydzienski**[2], **Barbara Pienkowska-Grela**[1], **Klaudia Borkowska**[1], **Jolanta Rygier**[1], **Aleksandra Kotyl**[1], **Natalia Malawska**[1], **Katarzyna Wojtkowska**[1], **Joanna Parada**[3], **Anita Borysiuk**[2], **Victor Murcia Pienkowski**[4], **Malgorzata Rydzanicz**[4], **Beata Grygalewicz**[1]

1 Cytogenetic Laboratory, Maria Sklodowska-Curie National Research Institute of Oncology, Warsaw, Poland, 2 Department of Pathology and Laboratory Diagnostics, Flow Cytometry Laboratory, Maria Sklodowska - Curie National Research Institute of Oncology, Warsaw, Poland, 3 Department of Cancer Biology, Maria Sklodowska-Curie National Research Institute of Oncology, Warsaw, Poland, 4 Department of Medical Genetics, Medical University of Warsaw, Warsaw, Poland

☯ These authors contributed equally to this work.
* renata.woroniecka@pib-nio.pl

**Data Availability Statement:** The datasets generated and/or analyzed during the current study are available from the European Nucleotide Archive

## Abstract

The occurrence of *MYC*-negative Burkitt lymphoma (BL) has been discussed for many years. The real frequency of the *MYC* insertion in *MYC*-negative BL is still unknown. Fine-needle aspiration biopsies of 108 consecutive patients with clinicopathologically suspected BL (suspBL) were evaluated by flow cytometry, classical cytogenetics, and fluorescence in situ hybridization (FISH). We found 12 cases (11%) without the *MYC* rearrangement by FISH with a MYC breakapart probe: two patients (1.9%) with cryptic *MYC/IGH* fusion (finally diagnosed as BL) and 10 patients (9.3%) with 11q gain/loss (finally diagnosed as Burkitt-like lymphoma with 11q aberration). The exact breakpoints of the cryptic *MYC/IGH* were investigated by next-generation sequencing. The *MYC* insertions' breakpoints were identified in *PVT1* in the first case, and 42 kb upstream of 5'*MYC* in the second case. To date, a molecular characterization of the *MYC* insertion in BL has only been reported in one case. Detailed descriptions of our *MYC* insertions in a routinely and consecutively diagnosed suspBL cohort will contribute to resolving the issue of *MYC* negativity in BL. In our opinion, the presence of the *MYC* insertions in BL and other lymphomas might be underestimated, because routine genetic diagnostics are usually based on FISH only, without karyotyping.

## Introduction

Burkitt lymphoma (BL) is a highly aggressive B-cell lymphoma and the fastest-growing human tumor type. The genetic hallmark of BL is *MYC* rearrangement (*MYC*R). This aberration is present in nearly all BL cases, mainly as a result of the chromosomal translocation t(8;14)(q24;q32), and less often due to the variant translocation t(2;8)(p11;q24) or t(8;22)(q24;q11) [1–3]. The molecular consequence of these translocations is the deregulated expression of the *MYC* oncogene. The overexpression arises as a result of the juxtaposition of *MYC* to the enhancer

(ENA) (http://www.ebi.ac.uk/ena/data/view/PRJEB42831).

**Funding:** This work was supported by the Maria Sklodowska-Curie National Research Institute of Oncology, Warsaw, Poland (https://www.pib-nio.pl/) Grant # SN/GW08/2020 (BG) and Count Jakub Potocki's Foundation, Warsaw, Poland (http://www.fpotockiego.org.pl/) Grant # 825/18 (BG). Some of the bioinformatic analyses presented in this study were performed on the ZEUS supercomputer located at the University of Science and Technology in Krakow, Poland. This research was supported in part by PLGrid Infrastructure (https://www.plgrid.pl/) Grant # ovcarnaseq (LMS). The funders had no role in study design, data collection and analysis, decision to publish, or preparation of the manuscript."

**Competing interests:** The authors have declared that no competing interests exist.

elements of one of the immunoglobulin (*IG*) genes: *IGH* (14q32), *IGK* (2p11), or *IGL* (22q11) [4]. Recent studies have described lymphomas, which morphologically and phenotypically resemble BL but have unique chromosome 11q aberrations (11q gain/loss) instead of *MYC*R. For these lymphomas, the term Burkitt-like lymphoma with 11q aberration (BLL,11q) was proposed as a new provisional entity in the revised 4th edition of the World Health Organization's WHO Classification of Tumors of Hematopoietic and Lymphoid Tissues [5]. Some rare cases with 11q gain/loss also have *MYC*R (BL,*MYC*R/11q) and are diagnosed as BL or high-grade B-cell lymphoma, not otherwise specified (HGBL,NOS), or even double-hit lymphoma (DHL) [6–8]. *MYC*R is also observed in other aggressive mature B-cell lymphomas (BCLs), such as HGBLs and diffuse large B-cell lymphomas (DLBCLs).

The breakpoints of the *MYC* are widely dispersed across the large >1 Mb region, and depend on the lymphoma subtype and translocation partner. In sporadic BL (sBL) with *MYC/IGH* fusion, breakpoints of the *MYC* are mapped within *MYC* or in close proximity to 5′*MYC* [9, 10]. On the other hand, the breakpoints of *MYC* involved in the variant translocations are located 16–350 kb from 3'*MYC* [9–13]. Most breakpoints within the *IGH* region in the *MYC/IGH* of sBL are located within switch regions (S), and only a minority, within the joining locus [14].

In almost all BLs, and other BCLs, the *MYC* fusion*s* are the results of karyotypically visible translocations. However, there are few data in the literature describing the *MYC* fusions arising from cryptic *MYC* insertion in different types of lymphomas [10, 15–19]. According to these data, the occurrence of the cryptic *MYC* insertion in BL is very rare, and only occasional cases of such insertions have been described. Detailed molecular characterizations of insertion breakpoints in BL are even more scarce—only one such case has been recorded [19].

Because the detection of every type of the *MYC*R is crucial for determining the final BL diagnosis, detailed knowledge regarding the molecular features and frequency of the *MYC* insertions in BL is very important. Herein, we present two cases of BL without typical, chromosomal *MYC* translocations and without 11q gain/loss out of 108 consecutive, mainly adult patients with BL/BLL,11q diagnosis. In these cases, with clinicopathological characteristics of classical BL, the karyotypically invisible insertion of *MYC* into the *IGH* locus and that of *IGH* into the *MYC* region were detected. We present a detailed characterization of these fusions on a molecular level obtained by next-generation sequencing (NGS). Thus, we confirm the rare occurrence of cryptic *MYC* fusions in BL patients with a frequency of 1.9% in patients with clinicopathologically suspected BL diagnosis (suspBL). We also discuss the significant role of the flow cytometry (FCM) evaluation of CD38 expression in establishing the final diagnosis of BL/BLL,11q and the value of karyotyping in distinguishing *MYC* insertions during routine BL diagnosis.

## Materials and methods

### Patients

The classical cytogenetics (CC) and/or fluorescence in situ hybridization (FISH) status of *MYC* was routinely analyzed in 108 consecutive patients with suspicion of BL, diagnosed at Maria Sklodowska-Curie National Research Institute of Oncology (Warsaw, Poland) from 2003 to 2020. In all patients with clinical or histopathology/immunohistochemistry (HP/IHC)-suspected BL diagnosis, fine needle aspiration biopsy (FNAB) for FCM/CC/FISH was performed. A diagnosis was established according to the 2016 revision of the WHO Classification of Lymphomas [5] and our practical FCM and IHC-based approach to the diagnosis of BL and BLL,11q [7]. All the BLL,11q cases diagnosed before the latest revision of the WHO classification were primarily diagnosed as *MYC*-negative BL and treated as *MYC*-negative BL at our

institute. Finally, *MYC*-positive BL was confirmed in 93 cases. *MYC*-negative BLL,11q and BL, *MYC*R/11q were established in 10 and 5 cases, respectively. Patients with HGBL,NOS with *MYC*R and DLBCL with *MYC*R were excluded because of different diagnoses after reviews of the HP slides, combined with FCM/IHC data and a more complex karyotype or clinical data obtained at follow-up.

## Classical cytogenetics

Cells prepared from the FNAB sample were fixed directly and cultured for 24 h without mitogen or for 48 or 72 h with DSP-30 (2 μM; TIBMolBiol, Berlin, Germany) together with IL-2 (200 U/mL; R&D Systems, Minneapolis, MN, USA). Chromosomes were stained with Wright for G,C-banding and analyzed using the MetaSystems Ikaros Imaging system (Metasysytems, Altlussheim, Germany), and the karyotypes were described according to the International System for Human Cytogenetic Nomenclature (ISCN 2016) [20].

## FISH

FISH analysis was performed on cultured cells in 104/108 patients. In 4/108 patients, a formalin-fixed paraffin- embedded (FFPE) tumors were used. In six patients both type of samples were used. FFPE specimens were prepared with a Pretreatment Reagent Kit (Vysis Abbott Molecular, Downers Grove, IL, USA) according to the manufacturer's protocol. For the routine diagnostics of patients with suspBL diagnoses, the following probes were used: BCL2 breakapart (BAP), BCL6 BAP, and MYC BAP (Vysis Abbott Molecular). For the precise evaluation of the *MYC* aberrations, the following probes were used: IGH BAP, IGH/MYC:CEP8 (Vysis Abbott Molecular), IGK/MYC, and IGL/MYC (CytoTest, Rockville, MD, USA). For the assessment of 11q gain/loss, the following panel of probes (11q gain/loss panel) was used: ATM SO, CCND1 SO, MLL BAP, TelVision 11q (D11S1037), and CEP11 (Vysis Abbott Molecular). The FISH results were analyzed using a fluorescence microscope, Axioskop2 (Carl Zeiss, Jena, Germany), documented by the ISIS Imaging System (Metasysytems, Altlussheim, Germany).

## Histopathology and immunohistochemistry

FFPE tissues were examined by standard HP/IHC, as described previously [7, 21] and characterized in S1 File. The IHC was performed using monoclonal antibodies specific for CD(3/5/ 10/20/38/43/ 44/56), BCL2, BCL6, LMO2, MYC, MUM1, Tdt, and Ki-67. Depending on the date of diagnosis, the panel of IHC varied but always included CD(3/10/20), BCL2, BCL6, and Ki-67. In the following years, the panel was expanded to include CD(5/38/43/44/56), MYC, LMO2, MUM1, and Tdt. Latent membrane protein 1 (LMP1) expression by IHC and Epstein–Barr virus (EBV) small nuclear RNA transcripts (EBER) by in situ hybridization (ISH) method was performed in some patients as described previously [7].

## Flow cytometry with cytological smears evaluation

The immunophenotype (CD38 PE-conjugated HB7 clone and other monoclonal antibodies) for the BL/BLL,11q diagnosis of cells obtained by the FNAB or ultrasound-guided FNAB of the lymph nodes/tumors was determined as previously described (see S1 File) [7, 21]. Antigen expression was quantified using FACSCalibur and FACSCanto II cytometers (Becton Dickinson Biosciences, San Jose, CA, USA) and was categorized according to the percentages of positive cells into three groups: '(–)'- no expression (<20% of neoplastic cells), '(+/–)'—expression in ≥20% but <100% of cells, and '(+)'—expression in 100% of lymphoma cells. Quantitative

expression was defined as (+)[higher] or (+)[comparable] than on control lymphocytes (i.e., CD38 (+)[higher] or CD38(+) in BL and BL,*MYC*R/11q or BLL,11q cells, respectively, compared to normal B- and T-lymphocytes). Simultaneously, cytological smears were stained with a hematoxylin-eosin (HE) and May-Grünwald-Giemsa stain for morphological evaluation, as described previously [7, 21].

## Next-generation sequencing

**DNA quality assessment.** Before the creation of the NGS libraries, the quality of the DNA was evaluated by real-time quantitative PCR using our personally developed method (see the detailed description in S1 File).

**Libraries and sequencing.** The NGS libraries used in this study were created, pooled, and enriched according to the SeqCap EZ HyperCap Workflow (Roche, Basel, Switzerland), using the NimbleDesign software to design the set of SeqCap EZ Choice hybridization probes (Roche) covering the following two regions in the human GRCh38 genome assembly: chr8:127,351,112–128,172,319 (*MYC*) and chr14:105,199,125–106,860,200 (*IGH*). The estimated coverage of the design equaled 92.8%. The obtained enriched multilibrary was then sequenced on a MiSeq next-generation sequencer (Illumina, San Diego, CA, USA) in paired-end mode ($2 \times 76$ bp).

**Bioinformatic analysis of the NGS results.** The quality of the FASTQ files was checked with the Fastqc app. Afterwards, the reads were mapped to the reference human genome (the GRCh38 assembly) with the HISAT2 aligner. The obtained BAM files were first analyzed with Qualimap [22] and Samtools [23] apps to determine the mapping quality and then subjected to deduplication with the MarkDuplicates program, a part of the Genome Analysis Toolkit (GATK) [24]. Finally, the deduplicated BAM files were used for the detection of intra- and interchromosomal translocations (CTX) with the CTX-explorer application (version:1.0), a personally developed piece of open-source software, available for download at https://github.com/lukszafron/CTX-explorer. In order to verify the sensitivity and specificity of the CTX-explorer-based breakpoint predictions, our results obtained with this piece of software were compared with the output of Breakdancer [25] and Delly [26], open-source programs developed by other research teams.

**In vitro verification of the CTX found in silico.** To verify the existence of interchromosomal translocations in the genomes, a unique pair of PCR primers for each patient was designed: Case 1 (forward primer: 5′–AGGAGCAACATAATGGGGGC–3′; reverse primer: 5′–CCTTTTCAGTTTCGGTCAGCC–3′); Case 2 (forward primer: 5′–GACGGTCAGCCACTTCTCTC–3′; reverse primer: 5′–GACTTGGACCTTGCCTGTCC–3′). The PCR and Sanger sequencing reactions were performed under conditions described in S1 File.

## Results

### Patients

Clinicopathological features and the results of HP/IHC revealed 108 patients with suspBL diagnosis from a total cohort of approximately 11,000 FCM/CC/FISH diagnoses of lymphomas obtained by FNAB material. This group of patients consisted of 102 adults with median age of 35 years (range, 19–79 years) and 6 children with median age of 8 years (range, 3–12 years). Among adult patients, 81 were male and 21 were female (ratio, 3.86:1). Among pediatric patients, 5 were male and 1 was female (ratio, 5:1). For the precise establishing of final diagnosis, FCM, CC and FISH of FNABs were performed (Table 1 and S1 Table). Both the CC and FISH were condacted in 86/108 patients. In the remaining 22/108 patients, FISH (20/108) or CC (2/108) were carried out. Some of the HP, FCM, molecular, and clinical data of these

**Table 1. The *MYC* and CD38 status with epidemiological data in 108 patients with suspected Burkitt lymphoma.**

| FISH + karyotype | Number of cases (% of cases) | FCM: CD38 | Final diagnosis | Age (years median, range) | Sex (male: female) |
|---|---|---|---|---|---|
| *MYC*R and/or t(8;V) | 91/108 (84%) | (+)higher | BL | 48 (3–68) | 3.47:1 |
| *MYC*R and t(8;V) and 11q gain/loss | 5/108 (4.7%) | (+)higher | BL,*MYC*R/11q | 31 (20–65) | 5:0 |
| *MYC*noR: | 12/108 (11%) | | | | |
| *MYC/IGH* | 2 (1.9%) | (+)higher | BL | 29 (22–36) | 1:1 |
| *MYC/IGL* | 0 | | | | |
| *MYC/IGK* | 0 | | | | |
| 11q gain/loss | 10 (9.3%) | (+)weaker | BLL,11q | 29 (20–79) | 10:0 |
| 11q gain/loss + *MYC/IGH* | 0 | | | | |
| 11q gain/loss + *MYC/IGL* | 0 | | | | |
| 11q gain/loss + *MYC/IGK* | 0 | | | | |

Abbreviations: FCM, flow cytometry; *MYC*R, the *MYC* rearrangement detected by MYC BAP probe; t(8;V), translocation of 8q24 (*MYC* locus) and one of the loci: 14q32 (*IGH*), 22q11 (*IGL*), and 2p11 (*IGK*); BL, Burkitt lymphoma; 11q gain/loss, duplication and deletion of 11q observed in karyotype and confirmed by FISH; BL, *MYC*R/11q, Burkitt lymphoma with both the *MYC* rearrangement and 11q gain/loss; *MYC*noR, lack of the *MYC* rearrangement detected by MYC BAP probe; BLL,11q, Burkitt-like lymphoma with 11q gain/loss.

patients have been published previously [6, 7, 21, 27]. Routine FISH analysis with MYC BAP, BCL2 BAP, and BCL6 BAP probes, performed in 106/108 patients, demonstrated a lack of *BCL2* and *BCL6* rearrangements in all cases and confirmed *MYC*R in 94/108 patients. In 2/108 patients (lack of FISH examination), *MYC*R was confirmed by a karyotype demonstrating the t (8;14)(q24;q32) translocation, for a total of 96/108 *MYC*R cases.

Among all the patients with *MYC*R (96/108), five had 11q gain/loss (5/108; 4.7%) (final BL, *MYC*R/11q diagnosis). In these patients, 11q gain was observed in the karyotype and further FISH examination, with an 11q gain/loss probe panel confirming the 11q aberration.

The remaining 12 patients demonstrated a lack of *MYC*R (12/108; 11%). All these patients were examined with the use of IGH/MYC, IGL/MYC, and IGK/MYC probes. In two patients (2/108), cryptic *MYC/IGH* fusions were confirmed (final BL diagnosis). In a further 10 patients, the karyotype and FISH with the 11q gain/loss probe panel revealed an 11q aberration (final BLL,11q diagnosis) (10/108; 9.3%). None of the patients without *MYC*R had cryptic *MYC/IGL* or *MYC/IGK* fusions.

The presence of two cases with *MYC* insertion among the patients with suspBL resulted in an *MYC* insertion frequency of 1.9% (2/108). Considering patients with final BL diagnosis, this frequency was 2% (2/93).

All the BL cases with just *MYC*R or the translocation of the 8q24 locus (91/108) were characterized by CD38(+)higher expression by the FNAB/FCM method. The BL,*MYC*R/11q cases (5/108) also demonstrated CD38(+)higher expression, while the expression of CD38 in BLL,11q cases (10/108) was significantly weaker—CD38(+). The BL cases without *MYC*R but with *MYC/IGH* fusion (2/108) (the *MYC* insertions described below) had CD38(+)higher expression. In both cases, despite the initial failure to confirm the *MYC*R, the FCM and HP/IHC results pointed to a BL diagnosis.

Some epidemiological data of patients with suspBL, including BL with *MYC* insertions as well as BLL,11q and BL,*MYC*R/11q are presented in Table 1.

## Case 1

**Clinical presentation, and pathomorphological and flow cytometry features.** A 22-year-old, HIV-negative Caucasian male presented with a bulky abdominal, extranodal tumor. His serum lactate dehydrogenase (LDH) (940 IU/L, $n < 240$), β2-microglobulin (4.44 ng/L, $n = 0.7$–$1.8$), d-dimer (1247 ng/mL, $n < 500$), C-reactive protein (CRP) (36.2 mg/L, $n < 5$ mg/L) and fibrinogen (3.59 g/L, $n = 1.7$–$3.5$) levels were elevated, with the biochemical features of renal failure, an ECOG performance status of 0, and Ann Arbor stage of IVA without B symptoms. Positron emission tomography (PET-CT) showed numerous extranodal lesions in the abdominal space and kidney. The patient has undergone a hemicolectomy and specimen from the tumor of the cecum revealed BL with a reduced number of apoptotic bodies and starry sky appearance in HP. IHC showed EBV-positive classic MYC-positive BL immunostaining for CD20+/CD10+/BCL6+/ BCL2–/MYC+ strong,100%/LMO2–/MUM1–/CD38 +strong/EBER+/EBV-LMP1–/CD43–/CD44–/CD56–/Ki-67 index > 98%/ CD3–/CD5–/TdT– (Fig 1). The FCM immunophenotype was determined three weeks after hemicolectomy on the recurrent abdominal tumor. BL cells were positive for CD45weaker/CD20bright/CD19bright/ CD22 (with the order according to median fluorescence intensity (MFI) being CD20 > CD19 > CD22)/CD10/CD38higher (with a MFI of 698 for CD38, compared to a MFI of 37 on T lymphocytes—Fig 2)/MYC/CD81higher/BCL6higher/ CD79β/HLA-DR/FMC7/CD43weaker/ CD49dweaker/CD52higher and surface immunoglobulin (IgM/κ), and negative for CD5/CD8/ CD11c/CD23/CD25/CD44/CD16&CD56/CD56/ CD138/CD200/CD305/BCL2/IgG/IgD/λ (Fig 3). The intracellular expression of MYC/BCL6 and a lack of BCL2 were detected after the permeabilization procedure. In addition, CD71 (+++) expression was detected in 100% of the cells. CD62L± and CD54± expression was weak in slightly over 20% of the tumor cells. The cytological smear stained with HE showed monomorphic medium-sized lymphoid cells with a small number of apoptotic bodies. A bone marrow (BM) HP/IHC was negative. The FCM confirmed the minimal cerebrospinal fluid (CSF) involvement of BL cells. The patient was treated with three courses of the R-CODOX-M/IVAC regimen (fractionated cyclophosphamide, doxorubicin, vincristine, and high-dose methotrexate alternating with fractionated ifosfamide, etoposide, and high-dose cytarabine, and triple-dose intrathecal therapy), leading to a complete response. Fifty-four months after diagnosis, the patient was still alive.

**Cytogenetics, FISH, and NGS.** CC and FISH with the MYC BAP probe of a recurrent abdominal biopsy demonstrated a normal karyotype 46,XY [20] and a lack of *MYC*R. FISH with the 11q gain/loss probe panel revealed normal results. However, subsequent FISH with IGH BAP and IGH/MYC dual fusion probes revealed the rearrangement of *IGH* and *IGH/ MYC* fusion via the insertion of the *IGH* into the *MYC* locus on a normal chromosome 8 (Fig 4). At the same time, FISH was performed on a FFPE abdominal tumor before recurrence, and we confirmed the *IGH/MYC* fusion.

The interchromosomal translocation analysis with our original CTX-explorer software (see Material and methods for details), followed by PCR and Sanger sequencing, showed that the breakpoint on chromosome 8 was located 158 kb downstream of 3′*MYC*, in the *PVT1* region (chr8:127,901,209) (Figs 4 and 5). Regarding the breakpoint on chromosome 14, it was between joining and constant *IGH* regions, 1.6 kb upstream of the Sμ switch region (ch14:105,862,125), according to the recently mapped *IGH* switch regions (Sμ: 105,856,501– 105,860,500) (S1A Fig) [28].

FISH with the IGH BAP probe revealed an atypical signal pattern (2Y1R), suggesting a breakpoint in the *IGH* region complementary to the 3′IGH BAP probe or duplication of this complementary region (Fig 4B). Considering the data regarding the exact location of the IGH BAP probe (according to the manufacturer's information), the *IGH* breakpoint was in the gap

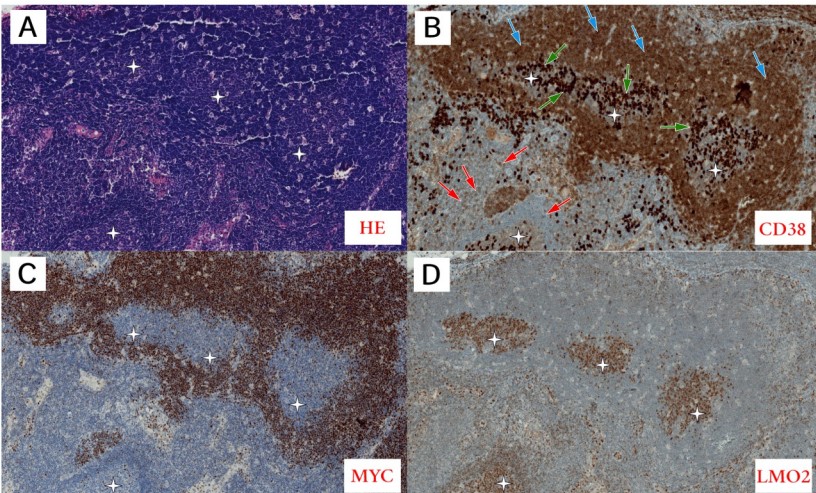

**Fig 1. Pathomorphological features of Case 1. A**: Pathomorphological features of cecal lymph node with partial involvement by Burkitt lymphoma (BL), obtained by surgery of the cecum. This image revealed BL with a reduced number of apoptotic bodies but with starry sky appearance without phagocytosis in histopathology as compared to the cecum tumor with the reduced number of apoptotic bodies and starry sky appearance. Diffuse growth is seen in terms of monomorphic medium-sized lymphoid cells showing a jigsaw puzzle effect of cytoplasmic borders. The round nuclei are similar in size and shape, showing open chromatin without clear nucleoli and with scanty amounts of cytoplasm (paraffin section stained with HE, original magnification, 40×). **B-D**: The other images show the immunophenotypic (IHC) features of BL in comparison to the part of the uninvolved lymph node (the asterisk indicates the unchanged germinal center (GC) of the lymph node). IHC showed classic MYC-positive BL immunostaining, CD38+$^{strong}$ (**B**)/MYC+ $^{strong,100\%}$ (**C**)/LMO2− (**D**) (original magnification 40×). The IHC test shows differences in CD38 staining between plasma cells (the strongest)(green arrows), BL cells (strong)(blue arrows), and T lymphocytes (the weakest, partially negative)(red arrows). On immunohistochemical staining, GC cells have weaker expression of CD38, with CD38+$^{higher}$ on plasma cells, no MYC, and a strong expression of LMO2 in most cells.

between the 3′IGH BAP probe and the 5′IGH BAP probe, which pointed to a duplication in the area complementary to the 3′IGH probe (S1C Fig).

## Case 2

**Clinical presentation, and pathomorphological and flow cytometry features.** A 35-year-old, HIV-negative Caucasian female presented with disseminated, mainly extranodal abdominal tumors, showing thickening of the stomach wall, an enlarged ovary, numerous lesions in the liver, ascites, and spinal canal infiltration with neurological symptoms and pain. Her serum LDH (2318 IU/L), β2-microglobulin (2.43 ng/L), d-dimer (1666 ng/mL), and fibrinogen (6.58 g/L) levels were elevated; ECOG performance status, 1; Ann Arbor stage, IVB with B symptoms. PET-CT showed numerous extranodal lesions in the abdominal space and nodal, massive BM involvement. BL cells from the peritoneal fluid and liver tumor were positive for CD45$^{weaker}$/CD20$^{bright}$/CD19$^{bright}$/CD22 (with an order according to MFI of CD20 > CD19 > CD22)/CD10/CD38$^{higher}$ (with an MFI of 873 for CD38, compared to an MFI of 36 on T lymphocytes)/CD81$^{higher}$/ CD79β/HLA-DR/CD43$^{weaker}$/ CD49d$^{weaker}$/ CD52$^{higher}$/CD54$^{higher}$/CD305/ MYC and surface immunoglobulin (IgD/IgM), while they were negative for CD5/CD8/CD11c/CD23/CD25/CD44/CD16&CD56/CD56/CD62L/CD200/ IgG/λ/κ and BCL6/. CD71 (+++) expression was detected in 100% of cells. FMC7(±) and BCL2(±)$^{weaker}$ expression was weak in slightly over 20% of the tumor cells (Fig 6). A monomorphic population of neoplastic lymphoid cells with a small number of apoptotic bodies in the background was visible in the cytological smear obtained from the peritoneal fluid and

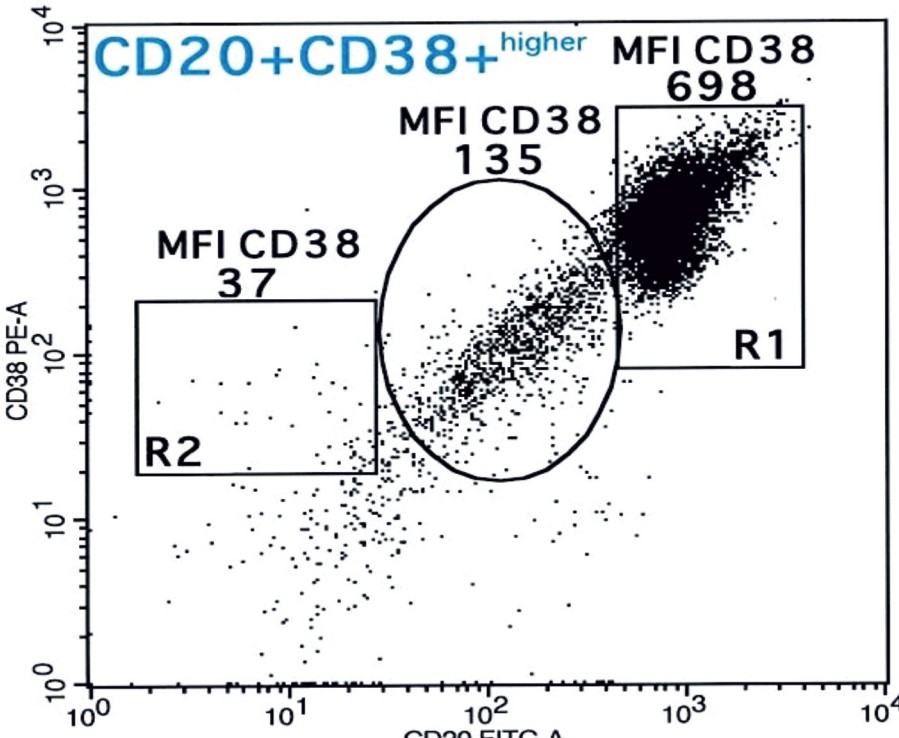

**Fig 2. Flow cytometry analysis of CD38 expression of Case 1.** Flow cytometry-based analysis of median fluorescence intensity (MFI) of CD38 expression on BL (698) in R1 was higher (CD38(+)$^{higher}$) than that for normal T-lymphocytes (37) in R2 and apoptotic bodies (135) (in a circle).

FNAB of the liver tumor (Fig 7A). A trephine biopsy revealed a diffuse proliferation of intermediate-sized atypical lymphoid cells with prominent central nucleoli, morphologically raising concern for BL, but also with a reduced number of apoptotic bodies and reduced starry sky appearance in HP (Fig 7B). The HP/IHC studies revealed that 90% of the BM involved BL cells. IHC showed classic MYC-positive BL immunostaining but still with BCL2(±)$^{weaker}$. A surgical biopsy from the stomach infiltrate revealed BL with the reduced number of apoptotic bodies and starry sky appearance in HP (Fig 7C). The IHC showed EBV-negative classic MYC-positive BL (but partial BCL2±$^{weaker}$ positive) immunostaining for CD20+/CD10 +/BCL6±/ MYC+$^{strong,100\%}$/ LMO2−/CD38+/ EBER−/EBV-LMP1−/MUM1−/CD43−/CD44 −/CD56−/Ki-67 index > 98%/CD3−/CD5−/ TdT− (Fig 7D). No CSF involvement by BL cells was confirmed by FCM. The patient was treated with four CODOX-M and IVAC alternating courses for patients with elevated risk. Thirty-nine months after diagnosis, the patient was still alive.

**Cytogenetics, FISH, and NGS.** The CC and FISH with a MYC BAP probe of peritoneum fluid cells demonstrated karyotype 46,XX,dup(1)(q21q42) [7]/46,idem,del(11)(q23) [6] and a lack of *MYC*R. However, subsequent FISH with an IGH/MYC dual fusion probe showed *MYC/IGH* fusion as an insertion of the *MYC* into the *IGH* locus on normal chromosome 14 (Fig 8).

The usage of the CTX-explorer app for identifying chromosomal breakpoints, followed by PCR and Sanger sequencing, revealed that the breakpoint on chromosome 8 was located 43 kb upstream of the 5′*MYC* (chr8:127,692,550), and the breakpoint on chromosome 14 was in a

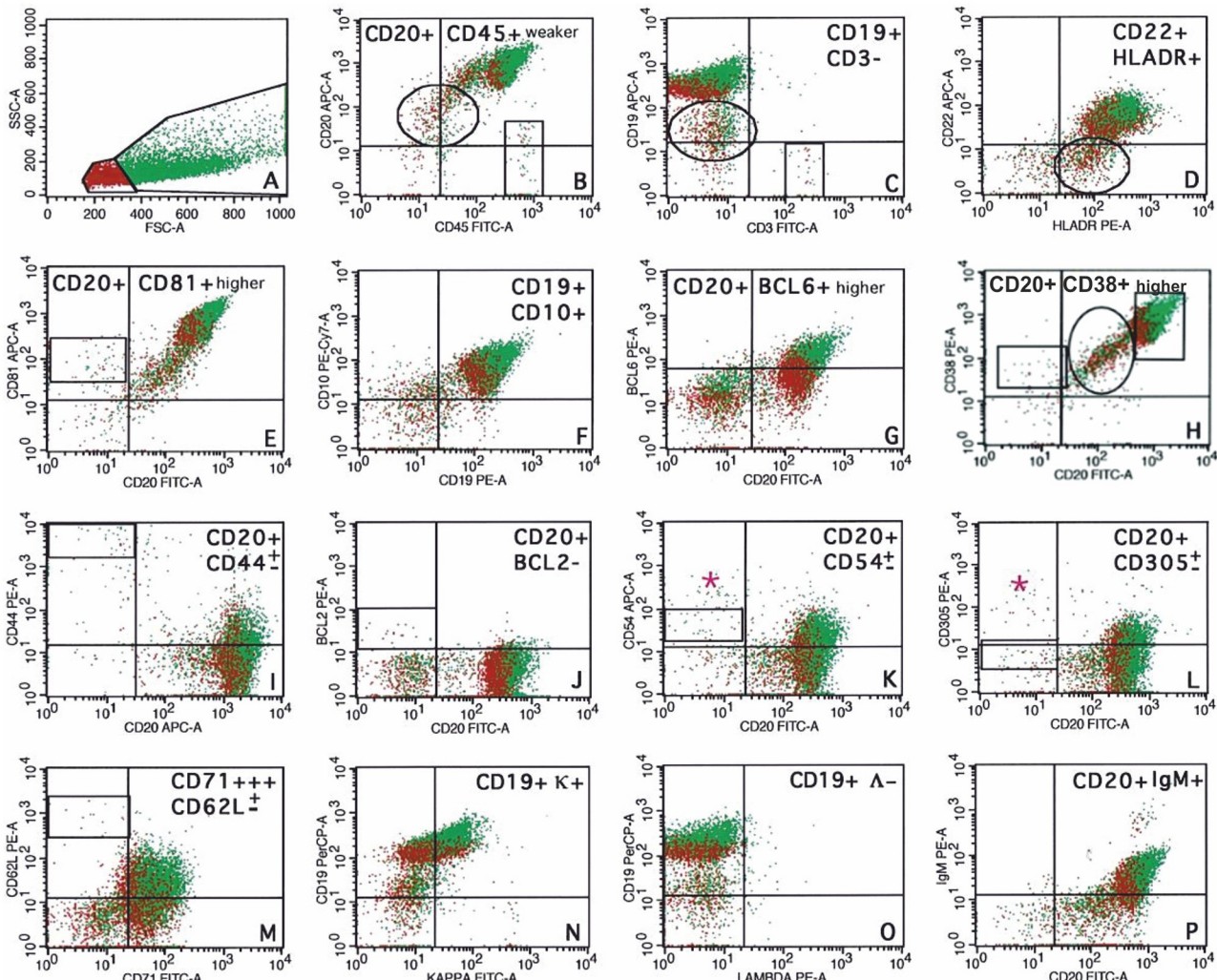

**Fig 3. Flow cytometry immunophenotyping of Case 1.** Fine-needle aspiration biopsy/flow cytometry analysis of BL. **A**: Forward scatter/side scatter dot plots present both small normal T lymphocytes (red cells) and usually larger lymphoma cells (green cells) with apoptotic bodies (marked by circles). BL expresses CD20/CD19/CD22 (MFI CD20 > CD19 > CD22) (**B–D**) as well as CD45+weaker/HLADR+. **E–I**: BL expresses a homogeneous phenotype of germinal center origin (CD81+higher/CD10+/BCL6+higher/CD38+higher/CD44±weaker (very low expression on a small subpopulation of cells)). **J–P**: BL expresses CD54±weaker/CD305±weaker/CD62L±weaker (very low expression on a small subpopulation of cells) but is negative for BCL2/lambda, with a restricted expression of IgM+ heavy/kappa+ light immunoglobulin chain. In addition, CD71+++ expression was detected in 100% of BL cells. Antigen expression of few macrophages and normal T-lymphocytes is marked with a pink asterisk and boxes, respectively. Dot-plots.

diversity *IGH* region, 2 kb downstream of 3′*IGHD2-2* (chr14:105,914,873) (Figs 5 and 8 and S1B Fig).

## CTX-explorer software for intra- and interchromosomal translocation detection

The greatest methodical problem in the accurate identification of *IG* translocations in NGS data is caused by the vast sequence diversity within the *IGH* due to somatic hypermutations and the genomic instability of malignant cells. These aspects significantly hamper the bioinformatic analysis of the NGS data. In order to increase the chance of inter-chromosomal translocation detection, we developed the CTX-explorer app, capable of identifying such genetic

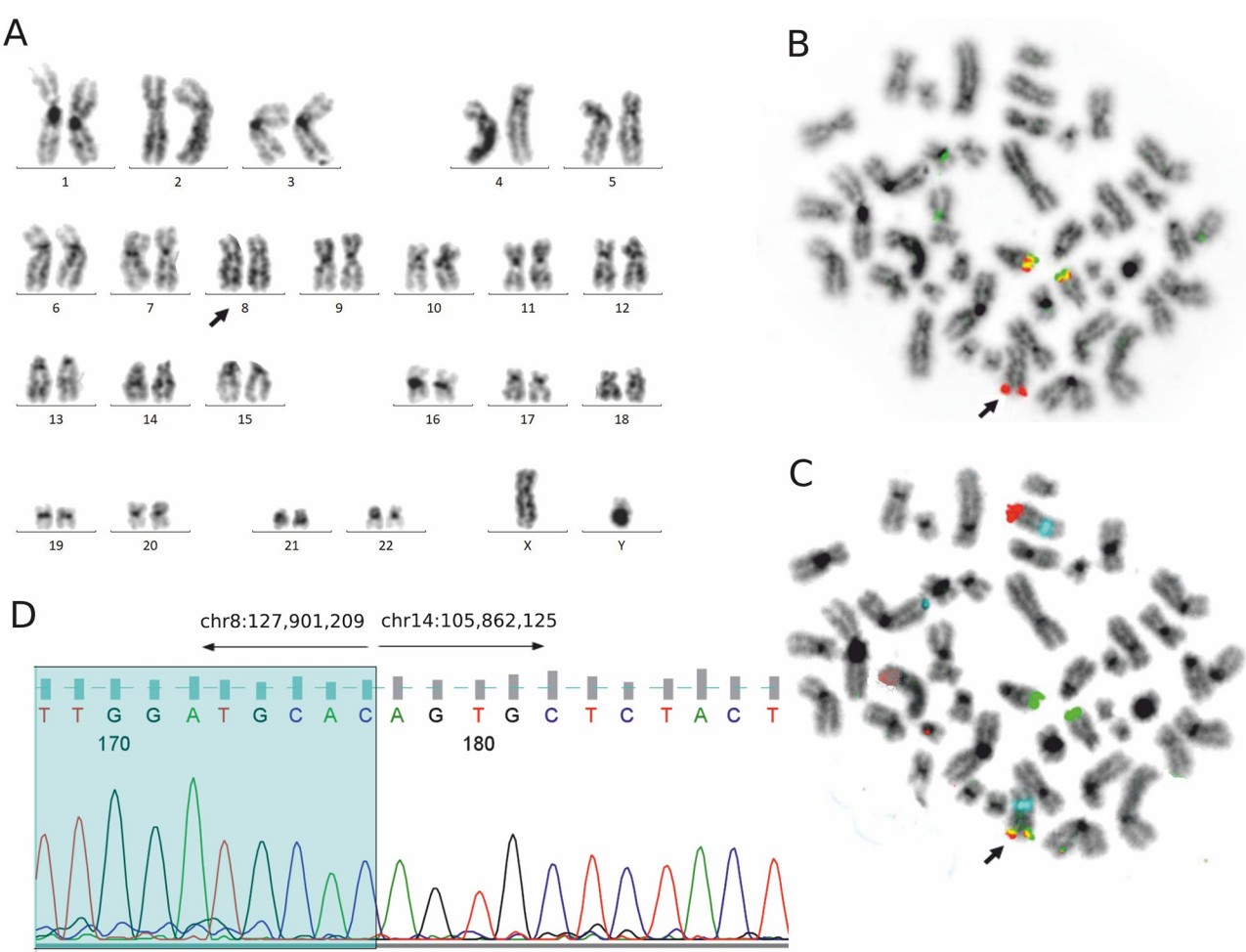

**Fig 4. Genetic findings in Case 1.** The thick black arrow indicates chromosome 8 with insertion of *IGH* and with *MYC/IGH* fusion. **A**: Karyotype 46, XY [20]. **B**: The same metaphase, FISH with IGH BAP probe: two non-rearranged IGH (yellow) signals on chromosomes 14 and one 3′IGH (red) signal on normal chromosome 8 indicating insertion of *IGH* into *MYC*. **C**: The same metaphase, FISH with IGH/MYC:CEP8 dual fusion probe: two centromere 8 (blue) signals on chromosomes 8, two IGH (green) signals on chromosomes 14, one MYC (red) signal on chromosome 8, and one MYC/IGH (yellow) signal on normal chromosome 8, indicating *MYC/IGH* fusion. **D**: Detailed breakpoints identified by PCR and Sanger sequencing: the break on chromosome 8 maps to the *PVT1* region; the break on chromosome 14 is located 1.6 kb upstream of the Sμ switch region.

alterations even in short NGS paired-end reads (75 bp or longer) with single-nucleotide precision. Notably, the mate pair reads are not required for this app to work. This feature is particularly advantageous if only poor-quality DNA (e.g., that extracted from FFPE samples) is available. The CTX-explorer program proved its usefulness, showing 100% specificity combined with an outstanding precision of detection—noticeably higher than that offered by other open-source apps (see Disccusion for details). In order to reduce the risk of CTX misidentification, either the gene set enrichment (as in this study) or exclusion of repetitive and low-complexity genome regions should be performed (with the windowmasker and dustmasker apps from the BLAST+ package [30]) before running the CTX-explorer software.

## Lymphoma *MYC* insertions reported in the literature

Table 2 presents 19 cases of *MYC* insertions in lymphomas previously reported in the literature [10, 15–19]. In all these cases, the *MYC* status was examined by FISH; in two cases, the

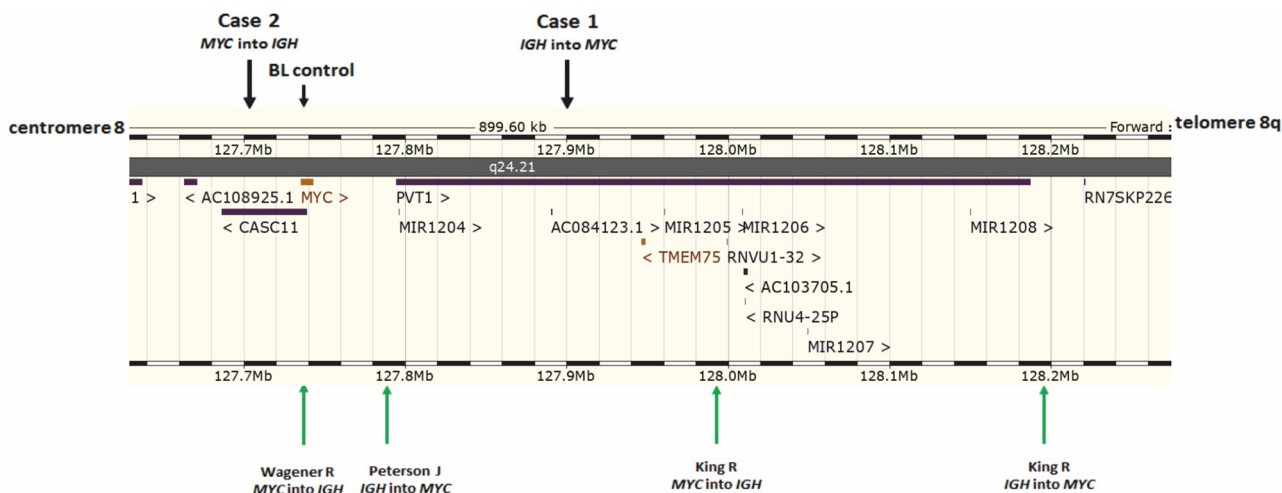

**Fig 5. Schematic view of the *MYC* insertion breakpoints in our data and previous literature.** In Case 1, the breakpoint on chromosome 8 was 158 kb downstream of 3′*MYC*, in the *PVT1* region. In Case 2, the breakpoint was 42 kb upstream of the 5′*MYC*. Green arrows indicate the *MYC* insertion breakpoints in lymphomas reported in the literature [17–19]. Visualization based on Ensembl 101: Aug 2020 [29]. BL control, BL without insertion, but with typical translocation t(8;14)(q24;q32) and with *MYC/IGH* fusion.

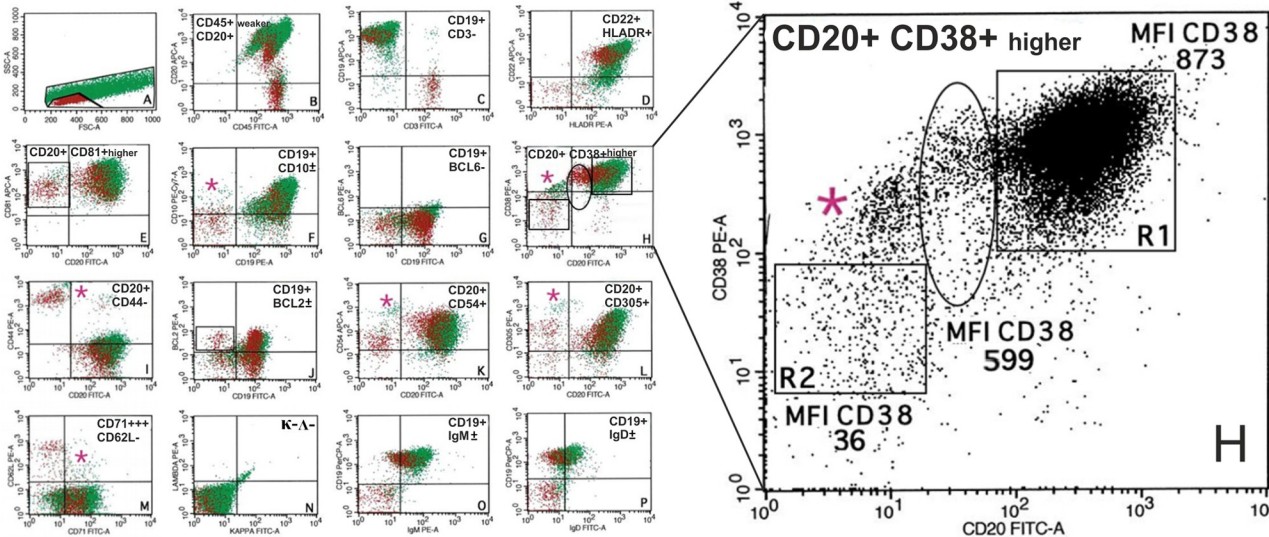

**Fig 6. Flow cytometry immunophenotyping including analysis of CD38 expression of Case 2.** FCM analysis of BL cells from the peritoneal fluid. **A**: Forward scatter/side scatter dot plots present both small normal T lymphocytes (red cells) and larger lymphoma cells (green cells) with a reduce number of apoptotic bodies (marked by circles). **B-D**: BL expresses: CD20/CD19/CD22 (with median fluorescence intensity (MFI) of CD20> CD19> CD22), as shown by monoclonal antibodies conjugated with the same fluorochrome, APC-A as well as CD45+weaker/HLADR+. **E-I**: BL expresses a homogeneous phenotype of germinal center origin (CD81+higher/CD10±/CD38+higher/CD44– but BCL6 negative. **H** (enlarged dot plot): FCM-based analysis of MFI of CD38 expression in BL. MFI of CD38 expression on BL (873) in R1 was higher (CD38(+)higher) compared to normal T-lymphocytes (36) in R2 and apoptotic bodies (599) (in a circle). **J-P**: BL expresses CD54+higher/CD305+higher/ BCL2±weaker (very low expression on a small subpopulation of cells) but is negative for CD62L/kappa/lambda with a restricted expression of IgM±/IgD±heavy immunoglobulin chain. In addition, CD71++ + expression is detected in 100% of BL cells. Antigen expression of few macrophages was marked with a pink asterisk (CD10/CD38/CD44/CD54/CD71/ CD305). Antigen expression of BL cells is compared to the expression on a subpopulation of normal T-lymphocytes (most antigens) (i.e. CD38/CD43/ CD44/CD45/CD54/CD81/BCL2) and on macrophages (i.e. CD54/ CD305) of the tumor and described as + higher for an antigen with a higher expression in BL cells compared to normal lymphocytes/ macrophages in 100% of cells; +, positive in 100% of BL cells; + weaker, for an antigen with a weaker expression than in lymphocytes/macrophages in 100% of cells; ± weaker, for an antigen with a weaker expression in BL cells compared to normal lymphocytes/macrophages in >20% to <100% of BL cells;−, no expression (i.e. expression in <20% BL cells). Dot-plots.

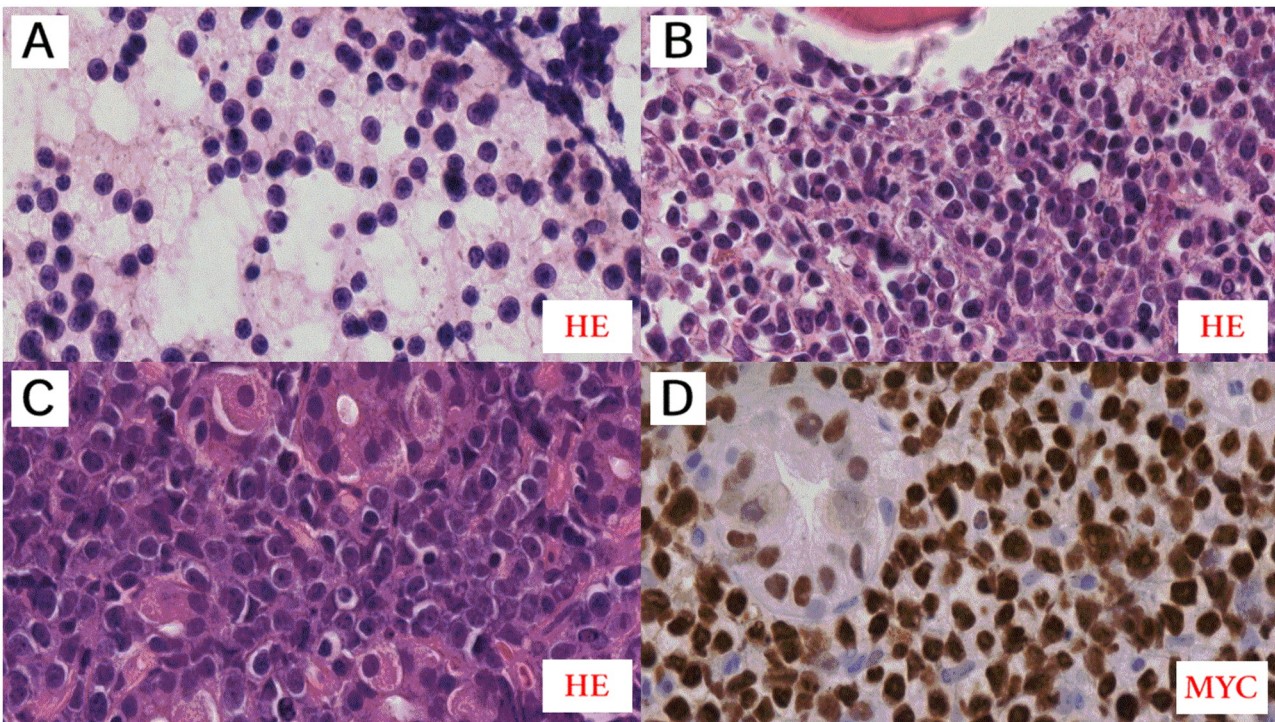

**Fig 7. Pathomorphological features of Case 2. A**: A monomorphic population of BL cells in the absence of apoptotic bodies in the background is visible in the cytological smear obtained from the FNAB of the liver tumor. Cytologic features with relatively uniform round nuclei, more cells with single, central nucleoli, and thin rims of cytoplasm—"small immunoblast" (cytological smear stained with HE, original magnification, 800×). **B**: A trephine biopsy showing heavy infiltration with BL. **C**: Gastric tissue biopsy showing heavy infiltration with BL. **B-C**: Both these images revealed BL with the reduced number of apoptotic bodies and starry sky appearance in HP. High magnification showing a "squaring off" of the cytoplasm. Also note the slight nuclear irregularity and more cells with single, central nucleoli (**B-C**: paraffin section stained with HE, original magnification, 800×). **D**: MYC protein immunostaining is strongly expressed by all BL cells. C-D: The images show stomach wall glands, which are also MYC positive (**D**) (**D**: original magnification 800×).

karyotype was also available. The type of insertion was defined in seven cases, revealing *MYC* inserted into *IGH* in four cases and *IGH* inserted into *MYC* in three cases. The diagnoses of all cases with *MYC* insertion were various and included BL, DHL, HGBL, DLBCL, primary cutaneous large B-cell lymphoma, leg type, mantle cell lymphoma, and plasma cell neoplasm. Detailed molecular analyses with the use of NGS were performed in four cases (Fig 5). In one HGBL reported by Peterson et al., a 200-kb fragment of *IGH* was inserted into *MYC*, upstream of and close to 5′*PVT1* [18]. King et al. described two cases of *MYC* insertion in BCL without precise diagnosis [17]. In the first case, *MYC* was inserted into *IGH* and the breakpoint was located in *PVT1*, 217 kb downstream of 3′*MYC*. In the second case, *IGH* was inserted into *MYC*, down-stream of and close to 5′*PVT1*. Wagener et al. presented a case of BL with the insertion of exons 2 and 3 of *MYC* into the I*GH* locus [19].

The *IGH* breakpoints were specified by Wagener et al. and Peterson et al. and were located within Sα1 and within both Sα2 and 3′ to Sμ, respectively. King et al. did not specify the break location in *IGH*, reporting that variable and diversity regions were affected.

## Discussion

The genetic hallmark of BL is a translocation of *MYC* and one of the *IG* genes. The real occurrence of BL without *MYC*R has been a subject of discussion for many years. Recently, some

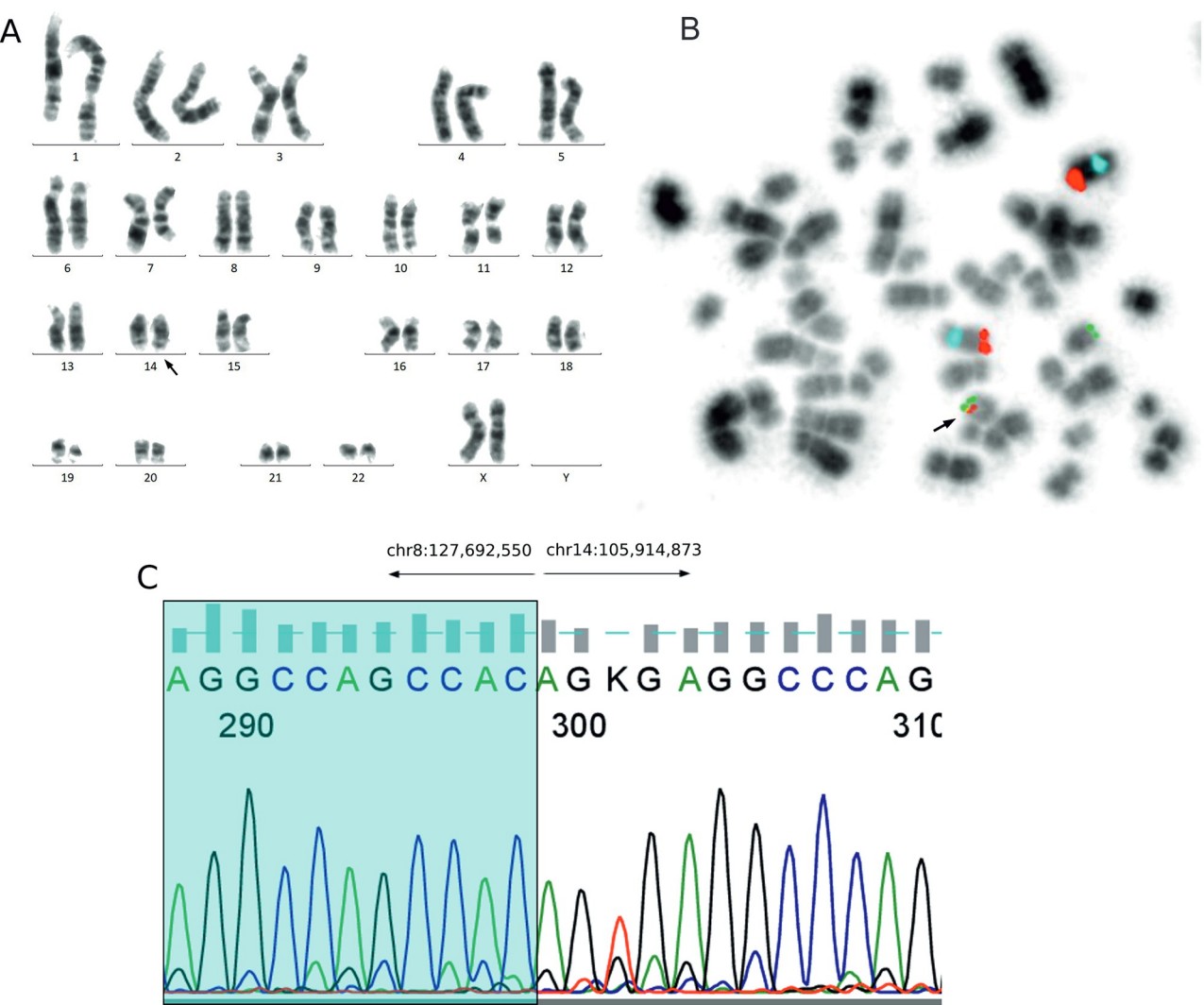

chr8:127,692,550   chr14:105,914,873

**Fig 8. Genetic findings in Case 2.** The thick black arrow indicates chromosome 14 with insertion of the *MYC* and with the *MYC/IGH* fusion. **A**: Karyotype 46,XX,dup(1)(q21q42) [7]/46,idem,del(11)(q23) [6]. **B**: Metaphase, FISH with IGH/MYC:CEP8 dual fusion probe: two centromere 8 (blue) signals on chromosomes 8, two MYC (red) signals on chromosomes 8, one IGH (green) signal on chromosome 14, and one MYC/IGH (yellow) signal on chromosome 14, indicating *MYC/IGH* fusion. **C**: Detailed breakpoints identified by PCR and Sanger sequencing: the break on chromosome 8 maps 43 kb upstream of the 5′*MYC*; the break on chromosome 14 is 2 kb downstream of 3′*IGHD2-2*.

*MYC*-negative BL cases have been described as having characteristic 11q gain/loss [31, 32]. The revised 4th edition of the WHO Classification of Lymphomas describes this entity as "Burkitt-like lymphoma with 11q aberration" [5]. On the other hand, some of the *MYC*-negative BL cases may still have *MYC*Rs, which appear as cryptic during testing by standard genetic methods. Since the presence of *MYC*R is crucial for establishing BL diagnosis, in rare *MYC*-negative BL cases, detailed examination is needed to determine the exact status of the *MYC* [8].

In our cohort of 108 cases with the clinicopathological features of BL, which accounted for approximately 1% (108/11,000) of all FNAB/FCM diagnosed lymphomas, we found 12 cases without *MYC*R as confirmed by CC and FISH with the MYC BAP probe. The one percent incidence of BL in our FNAB/FCM diagnosed cohort is in line with the incidence of sBL in Poland, in western Europe, and in the USA, where BL constitutes only 1–2% of all lymphomas

**Table 2. Review of the literature data regarding *MYC* insertion in lymphomas.**

| Authors | Number of cases | Type of insertion | Genetic methods | Diagnosis |
|---|---|---|---|---|
| Haralambieva et al., 2004 [10] | 1 | *MYC* into *IGH* | FISH | sBL (Caucasian) |
| May et al., 2010 [15] | 3 | *IGH* into *MYC* | FISH, classical cytogenetics | Suspected BL |
| | | *MYC* into *IGH* | FISH, classical cytogenetics | PCLBCL |
| | | one case—no data | FISH | HGBL |
| Sun et al., 2012 [16] | 11 | no data | FISH | Various diagnoses (BL, DLBCL, MCL, DHL) |
| Peterson et al., 2019 [18] | 1 | *IGH* into *MYC* | FISH, NGS (MPseq) | DHL |
| King et al., 2019 [17] | 2 | *IGH* into *MYC* | FISH, NGS (MPseq) | Various diagnoses (HGBL, DLBCL, PCN) |
| | | *MYC* into *IGH* | | |
| Wagener et al., 2020 [19] | 1 | *MYC* into *IGH* | FISH, NGS | BL |

Abbreviations: sBL, sporadic Burkitt lymphoma; BL, Burkitt lymphoma; PCLBCL, primary cutaneous large B-cell lymphoma, leg type; HGBL, high-grade B-cell lymphoma; DLBCL, diffuse large B-cell lymphoma; MCL, mantle cell lymphoma; DHL, double hit lymphoma; NGS, next-generation sequencing; MPseq, mate-pair sequencing; PCN, plasma cell neoplasm.

[5, 33]. Among our *MYC*-negative cases, 10 cases demonstrated 11q gain/loss, leading to a final diagnosis of BLL,11q. In the remaining two cases with final BL diagnosis, karyotypically cryptic *MYC/IGH* fusions were detected. In our report, BLL,11q were the majority of *MYC*-negative suspBL (83%) and accounted for approximately 9% of adult aggressive CD10(+) BCLs suspected of BL. For comparison, the studies in the literature reported the BLL,11q' incidence of 3 or 13% in suspBLs [32, 34]. On the other hand, we demonstrated that nearly 17% of the *MYC*-negative suspBL had cryptic *MYC* insertions. As we have previously emphasized, *MYC* negativity defined by using break apart probes and karyotyping does not exclude cryptic rearrangements, because both these methods cannot detect insertions of small chromosomal segments, which did not change the morphology of chromosomes. Considering these limitations, we have applied dual fusion probes to assess *MYC* status in our two *MYC*-negative cases, which did not have 11q gain/loss. Moreover, in all the cases with 11q aberrations and without *MYC*R, the status of *MYC/IGH*, *MYC/IGK*, and *MYC/IGL* fusions was also verified. None of these BLL,11q patients had a cryptic *MYC* insertion. This verification was necessary because, as we and others have described before [6–8], the occurrence of 11q gain/loss does not rule out *MYC*R. In this study, the concurrent presence of *MYC*R and 11q gain/loss was observed in approximately 5% of patients with suspBL. This information is worth underlining, because the data on the frequency of BL,*MYC*R/11q has not been published.

The literature data regarding the cryptic *MYC* insertions in lymphomas are scarce; only a few incidences of this aberration in BL have been reported (Table 2) [10, 15–19]. Moreover, a detailed molecular description of the cryptic *MYC/IGH* fusion breakpoints was given only in one case of BL, by Wagener et al. [19], and both breakpoints of *MYC* and *IGH* were typical of *MYC/IGH* in sBL. In our study, the *MYC* breakpoint in Case 2 was also typical of *IGH/MYC* fusions in sBL, which are mapped most often within *MYC* (in exon 1 and intron 1) or close to the 5′*MYC* [9, 13, 14]. However, in Case 1, the breakpoint resembled *MYC* breakpoints of variant *MYC/IGL* or *MYC/IGK* fusions in sBL, in which the *MYC* breakpoints are most often located more than 100 kb from the 3′*MYC*, in the *PVT1* [9, 11, 13, 14]. With respect to the *IGH* breakpoints, they were also only partially typical of the *MYC/IGH* fusions in sBLs. As described in the literature, most breakpoints within the *IGH* in sBLs with *MYC/IGH* fusions map to the switch and joining regions, and result from failed class switching (CSR) and VDJ recombinations, respectively [14, 35, 36]. In Case 1, the *IGH* break occurred outside but near the switch Sμ region, and errors in CSR may be the cause of this break. However, in Case 2, the

*IGH* break was 2 kb downstream of the *IGHD2-2*. The distance from the recombination signal sequences (RSS) region and the lack of N nucleotides at the breakpoint suggest that this insertion was not attributed to VDJ recombination [37].

It is worth mentioning that all the cryptic *MYC* insertions in lymphomas reported in the literature and in the present study result in the fusion of *MYC* with *IGH*. There are no data regarding cryptic fusions of *MYC* with *IGK* or *IGL*. The reason is that variant *MYC* fusions are less common than *MYC/IGH* fusions. The other cause is that *IGK/MYC* and *IGL/MYC* testing in *MYC*-negative lymphomas is performed sporadically. The additional emerging question is whether variant *MYC* fusions might occur in BLL,11q cases. In our report, we excluded variant fusions in BLL,11q cases, but further studies are needed.

Our results obtained with the CTX-explorer app were compared with the output of Breakdancer [25] and Delly [26], open-source programs developed by other research teams, the functionality of which included CTX detection. This comparison, performed for the two BL cases described herein and on two peripheral blood samples from healthy donors, revealed that CTX-explorer outperformed both competing apps in terms of the precision and specificity of the analysis. In Case 1, the breakpoint on chromosome 14 was identified perfectly by CTX-explorer (the results were identical to those obtained with the Sanger sequencing). On the contrary, Breakdancer missed the correct breakpoint by −284 bp, and Delly, by +1 bp. The breakpoint on chromosome 8 was misidentified by each application used (CTX-explorer: −23 bp, Breakdancer: −63 bp, and Delly: +5 bp). The detection of this breakpoint was tricky due to the low proportion of DNA molecules with this translocation, as assessed by examining the relevant BAM file with the Integrative Genomics Viewer (IGV). In Case 2, the breakpoint on chromosome 8 was identified by CTX-explorer; Delly made no mistake, while Breakdancer erred by +287 bp. On chromosome 14, each program made a mistake of −1 bp when trying to find the exact breakpoint. In fact, this error was caused by the HISAT2 aligner being unable to determine whether the last matching nucleotide was a part of the translocation or not (the same nucleotide was present on both chromosomes at the junction site). Finally, it is worth noting that the breakpoint detection with both CTX-explorer and Breakdancer was 100% specific, whereas the Delly app reported a t(8;14) translocation in one of two healthy blood donors.

The vast majority of reports regarding *MYC* insertions in lymphomas are based on retrospective analyses or isolated cases. In the present study, we describe two BL cases with *MYC* insertion, which were found during routine diagnostics for 108 patients with suspBL. At our institution, in all cases of clinically suspected BL/BLL,11q or HP/IHC-confirmed BL/BLL,11q, attempts are made to perform FNAB for further diagnostic tests [7, 27]. The high diagnostic accuracy and effectiveness of FNAB/FCM in BL/BLL,11q have been presented before [7]. According to these data, a lack of CD56 with CD38$^{higher}$ expression and CD56 expression without CD38$^{higher}$ proves to be a reliable, fast, easy, and cost-effective method for the estimation of *MYC*Rs and the 11q aberration in CD10(+) BCL, respectively. Moreover, FNAB samples enable us to culture cells for karyotyping, regardless of FISH. In the present report, all cases with *MYC*R as detected using the MYC BAP probe and without *MYC*R as detected using the MYC BAP probe, but with *MYC* insertions (BL and BL,*MYC*R/11q), were characterized by CD38(+)$^{higher}$ expression. On the other hand, the expression of CD38 in cases without *MYC*R (BLL,11q) was significantly weaker and comparable to CD38 expression in T cells. These data show that the overexpression of CD38 and *MYC*R detected using the MYC BAP probe allowed us to confirm *MYC*R in the vast majority of BL and BL,*MYC*R/11q cases, as well as to select cases for further examination of the *MYC* aberration type. In addition to FCM results, BL cases with the *MYC* insertion were characterized by the reduced number of apoptotic bodies and starry sky appearance in the histopathological examination, by strong MYC(+) expression

and a lack of LMO2(−) by IHC. Recently, such IHC:MYC(+)/LMO2(−) staining was found to be significantly associated with *MYC*R in CD10(+)BCL, including BL [7, 38], and consistent with low levels of *LMO2* expression in *MYC*-positive BL [39]. In both our *MYC*-negative cases with insertion, the false negative rate for the *MYC* BAP probe, comprehensively described by King et al. [17], with concomitant FCM/IHC results were enough for the use of MYC/IG probes. The MYC/IGH probe enabled us to detect the fusion in these cases; however, confirmation of the insertion was possible after CC and FISH on metaphases. Considering the significance of chromosome analysis in the detection of insertion, it is possible that the presence of the *MYC* insertions in lymphomas is undervalued because routine genetic diagnosis of suspBL in most laboratories is based on FISH only; karyotyping is rarely performed.

In summary, to the best of our knowledge, this is the largest study devoted to cryptic *MYC* insertions in consecutive mainly adult suspBL patients, routinely diagnosed by HP/IHC and FNAB/FCM/CC/FISH examinations at a single institution. We confirmed that cryptic *MYC* insertions in BL are extremely rare but not incidental. In our large group of patients clinico-pathologically suspected of BL, the frequency of this aberration was 1.9% and constituted 17% of *MYC*-negative suspBL. The remaining cases of *MYC*-negative suspBL were represented by BLL,11q. We detected the insertions through chromosome analyses and performed NGS examination of these alterations, which will extend our knowledge of the molecular features of very rare BL *MYC* insertions. The insertions we described were observed in sBL patients and resulted in cryptic *MYC/IGH* fusions. In one case, the breakpoint of the *MYC* was typical of *IGH/MYC* fusions in sBL, contrary to the other case in which the *MYC* break was as in variant *IG/MYC* fusions of sBL. Despite the rarity of *MYC* insertions, we believe that our study will substantially add to the understanding of *MYC*-negative BL and BLL,11q.

## Conclusions

The phenomenon of *MYC* insertions in lymphoma is known; however, data regarding the occurrence of this abnormality in BL are limited. Knowledge of the cryptic *MYC* insertion is important, particularly with respect to *MYC*-negative suspBL. We showed the molecular characteristics of insertion breakpoints in two sBL cases found in 108 consecutive patients with suspBL. *MYC* insertions constituted 17% of the *MYC*-negative group and 1.9% of the whole cohort. We expect that the appearance of the *MYC* insertions in lymphoma might be underestimated and that more studies on the frequency of this alteration in BL and BLL,11q are needed.

## Supporting information

**S1 Fig. Schematic view of *IGH* breakpoints in Cases 1 and 2 with *MYC* insertions.**
(PPTX)

**S1 Table. Summarized data of classical cytogenetic analyses in patients with suspected Burkitt lymphoma.**
(DOCX)

**S1 File. Methods.**
(DOCX)

## Author Contributions

**Conceptualization:** Renata Woroniecka, Grzegorz Rymkiewicz.

**Funding acquisition:** Lukasz M. Szafron, Beata Grygalewicz.

**Investigation:** Renata Woroniecka, Grzegorz Rymkiewicz, Lukasz M. Szafron.

**Methodology:** Renata Woroniecka, Lukasz M. Szafron, Laura A. Szafron, Joanna Parada, Victor Murcia Pienkowski, Malgorzata Rydzanicz.

**Resources:** Renata Woroniecka, Grzegorz Rymkiewicz, Katarzyna Blachnio, Zbigniew Bystydzienski, Barbara Pienkowska-Grela, Klaudia Borkowska, Jolanta Rygier, Aleksandra Kotyl, Natalia Malawska, Katarzyna Wojtkowska, Anita Borysiuk, Beata Grygalewicz.

**Software:** Lukasz M. Szafron.

**Visualization:** Zbigniew Bystydzienski.

**Writing – original draft:** Renata Woroniecka, Grzegorz Rymkiewicz, Lukasz M. Szafron.

**Writing – review & editing:** Renata Woroniecka, Grzegorz Rymkiewicz, Lukasz M. Szafron, Beata Grygalewicz.

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
