## [Decision Letter · Decision Letter 0]

27 Oct 2021

PONE-D-21-27800Cryptic *MYC* insertions in Burkitt Lymphoma: new data and a review of the literaturePLOS ONE

Dear Dr. WORONIECKA,

Thank you for submitting your manuscript to PLOS ONE. After careful consideration, we feel that it has merit but does not fully meet PLOS ONE’s publication criteria as it currently stands. Therefore, we invite you to submit a revised version of the manuscript that addresses the points raised during the review process.

We look forward to receiving your revised manuscript.

Kind regards,

Vincenzo L'Imperio

Academic Editor

PLOS ONE

Journal Requirements:

2.  In your Methods section, please provide additional information about the participant recruitment method and the demographic details of your participants. Please ensure you have provided sufficient details to replicate the analyses such as: a) a description of any inclusion/exclusion criteria that were applied to participant recruitment, b) a statement as to whether your sample can be considered representative of a larger population, c) a description of how participants were recruited, and d) descriptions of where participants were recruited and where the research took place.

3.  We note that Figure 3 in your submission contain copyrighted images. All PLOS content is published under the Creative Commons Attribution License (CC BY 4.0), which means that the manuscript, images, and Supporting Information files will be freely available online, and any third party is permitted to access, download, copy, distribute, and use these materials in any way, even commercially, with proper attribution. For more information, see our copyright guidelines: http://journals.plos.org/plosone/s/licenses-and-copyright.

    1.     You may seek permission from the original copyright holder of Figure(s) [#] to publish the content specifically under the CC BY 4.0 license.

Reviewers' comments:

Reviewer's Responses to Questions

**Comments to the Author**

1. Is the manuscript technically sound, and do the data support the conclusions?

Reviewer #1: Yes

Reviewer #2: Yes

2. Has the statistical analysis been performed appropriately and rigorously? 

Reviewer #1: Yes

Reviewer #2: N/A

3. Have the authors made all data underlying the findings in their manuscript fully available?

Reviewer #1: Yes

Reviewer #2: Yes

4. Is the manuscript presented in an intelligible fashion and written in standard English?

Reviewer #1: Yes

Reviewer #2: Yes

5. Review Comments to the Author

Reviewer #1: The article offers an overview of the molecular landscape of Burkitt lymphoma demonstrating, in a small - but not negligible - number of cases of BL (2%), the absence of MYC-R and 11q aberration with the concomitant presence of a cryptic MYC-IGH insertion, a situation whose existence was already known, also in other types of lymphoma, but having so far a low number of evidences in the literature.

The introduction and discussion are well done, while the results are sometimes difficult to understand due to the amount of abbreviations and numbers to be taken into account.

Main issues:

1) “Patients” paragraph is difficult to read, for example line 166 may be revised in this way “...demonstrating the t(8;14)(q24;q32) translocation, for a total of 96/108 MYCR cases.”

2) After all, it is not crystal clear the molecular reason why these cryptic insertions are not found with both karyotype analysis and MYC BAP probes, therefore it is necessary to insert an appropriate section into discussion.

Minor issues:

1) A paragraph dedicated to abbreviations must be added.

2) Line 28: double space after “FISH with”.

3) According to the literature, among suspBL how many are BL,11q? How many are BL,MYCR/11q?

4) Line 159: 108 suspBL diagnosis starting from a total cohort of how many cases?

5) Table 1: double space in column1-line2, after and/or.

6) In the tables all the abbreviations should have superscript and be sorted in the order of appearance.

7) Line 188: “All the BL cases with just MYCR or the translocation of the 8q24 locus (n = 91) without 11q aberration were characterized…”

8) Line 202: in my opinion, it should be said here that the patient has undergone a hemicolectomy.

9) Figure 1: each square of the histological image should have its letter (1 a.b.c.d), while figure 1b and 1c should become figure 2 and 3. Doing so, line 230 clarifications are no longer necessary (“upper row” and so on). The EE image has a low quality. An inset with an evident blastoid morphology should be added. The CD38 image could benefit from pointers or arrows to focus on plasma cells, BL cells and T lymphocytes.

10) Figure 4b has a low quality.

11) Lines 456-467: this part should be integrated into conclusions.

12) Supporting methods, immunohistochemistry: “...antigen-retrieval technique WERE...”

13) Supporting methods, flow cytometry: the sentence “Four to ten separate needle passes…” is potentially confusing. Maybe it’s better to say “Within the context of a single FNAB, four to ten separated needle passes…”.

14) S1 table: column1-line2 should be named “Total karyotype analysis”.

Reviewer #2: This research article focuses on a detailed description of two cases clinicopathologically suspected for Burkitt lymphoma (BL) but negative by FISH for typical chromosomal MYC translocations and without 11q gain / loss. In particular, a series of cases clinicopathologically suspected for BL was evaluated by flow-cytometry, classical cytogenetics (CC) and FISH. Of these, 12 cases were found to be negative for MYC rearrangements by FISH; subsequent karyotyping and FISH analyzes revealed the presence in 10 cases of 11q gain / loss and in 2 cases of cryptic MYC / IGH fusion. Considering the latter, Next Generation Sequencing (NGS) studies were then performed for a precise genetic and molecular characterization of these rare fusions and the results obtained were compared with the rare data currently available in the literature.

This study also describes the diagnostic workflow in cases suspicious for BL, starting from the clinicopathological evaluation and integrating various and different methods (flow cytometry, immunohistochemistry, CC, NGS), showing how each of them can be of fundamental and complementary importance in the diagnostic process.

The study is very well constructed, detailed and exposed in all its parts, also at the level of supplementary materials and methods; in addition, it addresses a topic (cryptic MYC insertions in BL) of fundamental importance but still with little data reported in the literature.

MAJOR ISSUES

- In both clinical cases (case 1 and case 2), the age of the patients, the ethnicity and the HIV status were reported while the EBV status lacks. It would be important to know if it has been determined or not, in order to be able to correlate the data obtained from the study with this data as well.

-Unlike case 1, in case 2 clinical presentation and pathomorphological features are only described in the text. You should add some pictures (histological, cytological, radiological) as well.

-Figure 4B: it is very difficult to distinguish the FISH signals reported in the description. You should provide an image with better definition.

MINOR ISSUES

-Page 4, lines 85-88: you should report the part concerning the re-definition of “BLL, 11q” cases diagnosed before the WHO review in 2016 in the results section.

-Page 5, line 99: you should define the exact number of cases in which FISH was performed on cultured cells and the exact number of cases in which FISH was performed on formalin-fixed paraffin-embedded (FFPE) sections.

-Page 5, line 117: you should modify the title of the paragraph, also including the fact that a morphological evaluation on cytological smears was performed.

-Page 7, lines 151-156: in “Materials and methods” it is reported that PCR and Sanger sequencing reactions were performed to verify the existence of the two interchromosomal translocations detected by NGS. You should report the results obtained in the results section.

-Page 7, line 161: you should express in brackets the exact number of cases in which FISH was performed and the exact number of cases in which classical cytogenetic was performed.

-Page 10, line 198: you should also express the extended version of the “CRP” acronym.

-Page 14, lines 294-295: immunohistochemical evaluation of BCL6 is reported twice. Regarding BCL6 results reported in the text, you should also check the agreement with the same data reported in the diagrams of figure S2B.

-Pages 19-20, lines 411-428: you should add that this analysis was performed in the “Materials and methods” section too.

---

## [Author Response · Author response to Decision Letter 0]

3 Dec 2021

Dear Prof. Vincenzo L’Imperio,

Thank you for giving us the opportunity to submit a revised draft of our manuscript titled “Cryptic MYC insertions in Burkitt Lymphoma: new data and a review of the literature” to PLOS ONE. We appreciate the time and effort that you and the reviewers have dedicated to providing your valuable feedback on our manuscript. We are grateful to the reviewers for their critical, insightful and very constructive comments on our paper. We believe that the revised version of our paper addresses all concerns by the referees in detail, because we incorporated changes to reflect most of the suggestions provided by the reviewers. 

We have highlighted the changes within the manuscript (file “Revised Manuscript with Track Changes”).

Here is a point-by-point response to the reviewers’ comments and concerns.

Comments from Journal:

Comment 1. Please ensure that your manuscript meets PLOS ONE's style requirements, including those for file naming. The PLOS ONE style templates can be found at

Response: Thank you for reminding us about manuscript and file requirements. We have done our best to meet these requirements.

Comment 2. In your Methods section, please provide additional information about the participant recruitment method and the demographic details of your participants. Please ensure you have provided sufficient details to replicate the analyses such as: a) a description of any inclusion/exclusion criteria that were applied to participant recruitment, b) a statement as to whether your sample can be considered representative of a larger population, c) a description of how participants were recruited, and d) descriptions of where participants were recruited and where the research took place.

Response: Thank you for pointing this out. We have added demographic and recruitment details in Materials and methods section, Results section and Discussion section of the revised manuscript:

Materials and methods (page 4, lines 81-86): “The classical cytogenetics (CC) and/or fluorescence in situ hybridization (FISH) status of MYC was routinely analyzed in 108 consecutive adult patients with suspicion of BL, diagnosed at Maria Sklodowska-Curie National Research Institute of Oncology (Warsaw, Poland) from 2003 to 2020. This group of patients consisted of 102 adults with median age of 35 years (range, 19-79 years) and 6 children with median age of 8 years (range, 3-12 years). Among adult patients, 81 were male and 21 were female (ratio, 3.86:1). Among pediatric patients, 5 were male and 1 was female (ratio, 5:1).”

Results (page 8, lines 170-172): “Clinicopathological features and the results of HP/IHC revealed 108 patients with suspBL diagnosis from a total cohort of approximately 11,000 FCM/CC/FISH diagnoses of lymphomas obtained by FNAB material.”

Results, Table 1: new columns with age and sex:

FISH + karyotype Number of cases

(% of cases) FCM: CD38 Final diagnosis Age 

(years median, range)

 Sex 

(male: female)

MYCR and/or t(8;V) 91/108 (84%) (+)higher BL 48 (3-68) 3.47:1

MYCR and t(8;V) and 11q gain/loss 5/108 (4.7%) (+)higher BL,MYCR/11q 31 (20-65) 5:0

MYCnoR : 12/108 (11%) 

 MYC/IGH 2 (1.9%) (+)higher BL 29 (22-36) 1:1

 MYC/IGL 0 

 MYC/IGK 0 

 11q gain/loss 10 (9.3%) (+)weaker BLL,11q 29 (20-79) 10:0

 11q gain/loss + MYC/IGH 0 

 11q gain/loss + MYC/IGL 0 

 11q gain/loss + MYC/IGK 0 

Results (page 10, lines 208-209): ” Some epidemiological data of patients with suspBL, including BL with MYC insertions as well as BLL,11q and BL,MYCR/11q are presented in Table 1.”

Discussion (page 20, lines 431-435): “In our cohort of 108 cases with the clinicopathological features of BL, which accounted for approximately 1% (108/11,000) of all FNAB/FCM diagnosed lymphomas, we found 12 cases without MYCR confirmed by CC and FISH with the MYC BAP probe. The one percent incidence of BL in our FNAB/FCM diagnosed cohort is in line with the incidence of sBL in Poland, in western Europe, and in the USA, where BL constitutes only 1-2% of all lymphomas [5,33].”

Comment 3. We note that Figure 3 in your submission contain copyrighted images. All PLOS content is published under the Creative Commons Attribution License (CC BY 4.0), which means that the manuscript, images, and Supporting Information files will be freely available online, and any third party is permitted to access, download, copy, distribute, and use these materials in any way, even commercially, with proper attribution. For more information, see our copyright guidelines: http://journals.plos.org/plosone/s/licenses-and-copyright.

Response: Thank you for pointing this out. As a permission for publication of Figure 3 (Fig 5 in the revised version of the manuscript), we have submitted a file with Ensembl answer for permission request. According to this email and information on Ensembl website, there is no restriction on re-use of Ensembl images. According to Ensembl re-use policy, we have added relevant information in the text of manuscript: 

a. We have labeled all modified Ensembl images (Fig 5 and S1 Fig)

b. We have cited the Ensembl release I retrieved my data

c. We have added citing Ensembl, with the most recent overview article (References, no.29):

Fig 5. “Visualization based on Ensembl 101: Aug 2020 [29].”

S1 Fig. “Visualization based on Ensembl 101: Aug 2020 [Howe KL, et al. Ensembl 2021. Nucleic Acids Res. 2021; 49(1): 884–891. doi:10.1093/nar/gkaa942]”

29. Howe KL, Achuthan P, Allen J, Allen J, Alvarez-Jarreta J, Amode MR, et al. Ensembl 2021.

Nucleic Acids Res. 2021; 49(1): 884–891. doi:10.1093/nar/gkaa942.

Comment 4: Please review your reference list to ensure that it is complete and correct. If you have cited papers that have been retracted, please include the rationale for doing so in the manuscript text, or remove these references and replace them with relevant current references. Any changes to the reference list should be mentioned in the rebuttal letter that accompanies your revised manuscript. If you need to cite a retracted article, indicate the article’s retracted status in the References list and also include a citation and full reference for the retraction notice.

Response: We thank for pointing this out. We have reviewed the reference list. The remarks are given below:

a) According to reviewers comments, we have added relevant information and we have to add three papers in reference list: 

No. 29. Howe KL, et al. Ensembl 2021. Nucleic Acids Res. 2021; 49(1): 884–891. doi:10.1093/nar/gkaa942

No. 32. Szumera-Ciećkiewicz A , et al. Distribution of lymphomas in Poland according to World Health Organization classification: analysis of 11718 cases from National Histopathological Lymphoma Register project - the Polish Lymphoma Research Group study. Int J Clin Exp Pathol. 2014; 15: 3280-3286. PMID: 25031749

No. 34. Au-Yeung RKH, et al. Experience with provisional WHO-entities large B-cell lymphoma with IRF4-rearrangement and Burkitt-like lymphoma with 11q aberration in paediatric patients of the NHL-BFM group. Br J Haematol. 2020; 190(5):753-763. doi: 10.1111/bjh.16578. 

b) We have change the order of two papers, because we have added some information in Materials and Methods section (according to reviewers comments):

 No. 25. (previously No. 33) Chen K, et al. BreakDancer: An algorithm for high-resolution mapping of genomic structural variation. Nat Methods. 2009; 6: 677–681. doi:10.1038/nmeth.1363.

 No. 26 (previously no. 34) Rausch T, et al. DELLY: Structural variant discovery by integrated paired-end and split-read analysis. Bioinformatics. 2012; 28: i333–i339. doi:10.1093/bioinformatics/bts378.

c) Shiramizu B, Barriga F, Neequaye J, Jafri A, Dalla-Favera R, Neri A, et al. Patterns of Chromosomal Breakpoint Locations in Burkitt’s Lymphoma: Relevance to Geography and Epstein-Barr Virus Association. Blood. 1991; 77: 1516–1526. doi:10.1182/blood.V77.7.1516.1516.

In database Pubmed, this paper has PMID:1849033, but it does not have DOI number. The abovementioned DOI number functions on the web browser.

d) Sun G, Montella L, Yang M. MYC Gene FISH Testing in Aggressive B-Cell Lymphomas: Atypical Rearrangements May Result in Underreporting of Positive Cases. Blood. 2012; 120: 1552. doi:10.1182/blood.v120.21.1552.1552.

This paper does not have any data in Pubmed, but the abovementioned DOI number functions on the web browser, in ASH publications (https://ashpublications.org/blood/article/120/21/1552/103122/MYC-Gene-FISH-Testing-in-Aggressive-B-Cell ).

e) Rymkiewicz G, Zajdel M, Paziewska A, Blachnio K, Grygalewicz B, Woroniecka R, et al. Molecular analyses and an innovative diagnostic algorithm in MYC-negative Burkitt-like lymphoma with 11q aberration: A single institution experience. Hematol Oncol. 2019; 37: 190–190. doi:10.1002/hon.4_2630.

This paper does not have any data in Pubmed, but the abovementioned DOI number functions on the web browser (https://onlinelibrary.wiley.com/doi/full/10.1002/hon.4_2630 ).

Comments from Reviewer #1:

MAIN ISSUES

Comment 1: “Patients” paragraph is difficult to read, for example line 166 may be revised in this way “...demonstrating the t(8;14)(q24;q32) translocation, for a total of 96/108 MYCR cases.”

Response: We apologize for difficulties in reading this paragraph. According to the Reviewer's suggestion, the paragraph Patients in Results section has been changed in the revised manuscript as requested (page 8, lines 173-179, 202-207):

“Both the CC and FISH were made in 86/108 patients. In the remaining 22/108 patients, FISH (20/108) or CC (2/108) were carried out. Some of the HP, FCM, molecular, and clinical data of these patients have been published previously [6,7,21,27]. Routine FISH analysis with MYC BAP, BCL2 BAP, and BCL6 BAP probes, performed in 106/108 patients, demonstrated a lack of BCL2 and BCL6 rearrangements in all cases and confirmed MYCR in 94/108 patients. In 2/108 patients (lack of FISH examination), MYCR was confirmed by a karyotype demonstrating the t(8;14)(q24;q32) translocation, for a total of 96/108 MYCR cases.”

“All the BL cases with just MYCR or the translocation of the 8q24 locus (91/108) were characterized by CD38(+)higher expression by the FNAB/FCM method. The BL,MYCR/11q cases (5/108) also demonstrated CD38(+)higher expression, while the expression of CD38 in BLL,11q cases (10/108) was significantly weaker - CD38(+). The BL cases without MYCR but with MYC/IGH fusion (2/108) (the MYC insertions described below) had CD38(+)higher expression. In both cases, despite the initial failure to confirm the MYCR, the FCM and HP/IHC results pointed to a BL diagnosis.”

Comment 2: After all, it is not crystal clear the molecular reason why these cryptic insertions are not found with both karyotype analysis and MYC BAP probes, therefore it is necessary to insert an appropriate section into discussion.

Response: We agree with this and appropriate explanation was added in the Discussion section of the revised manuscript as requested (page 20, lines 441-446):

„As we have previously emphasized, MYC negativity defined by using break apart probes and karyotyping does not exclude cryptic rearrangements, because both these methods cannot detect insertions of small chromosomal segments, which did not change the morphology of chromosomes. Considering these limitations, we have applied dual fusion probes to assess MYC status in our two MYC-negative cases, which did not have 11q gain/loss.”

Minor issues:

Comment 1: A paragraph dedicated to abbreviations must be added.

Response: We appreciate the reviewer’s suggestion, but after consulting the Editor, in our case it should be enough to spell the terms at least the first time they have been used along the text . 

Comment 2: Line 28: double space after “FISH with”.

Response: Thank you for pointing this out, we have deleted double space (page 2, line 29 in the revised manuscript).

Comment 3: According to the literature, among suspBL how many are BL,11q? How many are BL,MYCR/11q?

Response: We agree with this and we have incorporated your suggestion in Discussion section of the revised manuscript (page 20, lines 439-440, 451-453) : 

“For comparison, the studies in the literature reported the BLL,11q’ incidence of 3 or 13% in suspBLs [32,34].”

“This information is worth underlining, because the data on the frequency of BL,MYCR/11q has not been published.”

Comment 4: Line 159: 108 suspBL diagnosis starting from a total cohort of how many cases?

Response: Thank you for this suggestion. We have supplemented text in Results section of the revised manuscript as requested (Patients paragraph, page 8, lines 170-172): 

“Clinicopathological features and the results of HP/IHC revealed 108 patients with suspBL diagnosis from a total cohort of approximately 11,000 FCM/CC/FISH diagnoses of lymphomas obtained by FNAB material.”

Additionally, the obtained result was included in Discussion section of the revised manuscript as requested (page 20, lines 431-435): 

“In our cohort of 108 cases with the clinicopathological features of BL, which accounted for approximately 1% (108/11,000) of all FNAB/FCM diagnosed lymphomas, we found 12 cases without MYCR confirmed by CC and FISH with the MYC BAP probe. The one percent incidence of BL in our FNAB/FCM diagnosed cohort is in line with the incidence of sBL in Poland, in western Europe, and in the USA, where BL constitutes only 1-2% of all lymphomas [5,33].”

Comment 5: Table 1: double space in column1-line2, after and/or.

Response: Thank you for pointing this out, we have deleted double space.

Comment 6: In the tables all the abbreviations should have superscript and be sorted in the order of appearance.

Response: Thank you for pointing this out. We have changed the order of abbreviations (we have sorted them in the order of appearance) and we have revised them. However, we did not add the superscripts to all the abbreviations, because we think, that superscripts in all abbreviations will hamper reading. To keep this format, we deleted two superscripts:

Table 1 abbreviations: “FCM, flow cytometry; MYCR, the MYC rearrangement detected by MYC BAP probe; t(8;V), translocation of 8q24 (MYC locus) and one of the loci: 14q32 (IGH), 22q11 (IGL), and 2p11 (IGK); BL, Burkitt lymphoma; 11q gain/loss, duplication and deletion of 11q observed in karyotype and confirmed by FISH; BL,MYCR/11q, Burkitt lymphoma with both the MYC rearrangement and 11q gain/loss; MYCnoR, lack of the MYC rearrangement detected by MYC BAP probe; BLL,11q, Burkitt-like lymphoma with 11q gain/loss.”

Table 2 abbreviations: “sBL, sporadic Burkitt lymphoma; BL, Burkitt lymphoma; PCLBCL, primary cutaneous large B-cell lymphoma, leg type; HGBL, high-grade B-cell lymphoma; DLBCL, diffuse large B-cell lymphoma; MCL, mantle cell lymphoma; DHL, double hit lymphoma; NGS, next-generation sequencing; MPseq, mate-pair sequencing; PCN, plasma cell neoplasm.”

Comment 7: Line 188: “All the BL cases with just MYCR or the translocation of the 8q24 locus (n = 91) without 11q aberration were characterized…”

Thank you for this suggestion. We have added “just” in Results section of the revised manuscript as requested (Patients paragraph, page 10, lines 202-203): 

“All the BL cases with just MYCR or the translocation of the 8q24 locus (91/108) were characterized by CD38(+)higher expression by the FNAB/FCM method.”

Comment 8: Line 202: in my opinion, it should be said here that the patient has undergone a hemicolectomy.

Response: Thank you for pointing this out. We have supplemented text in the “description of Case 1” of the revised manuscript as requested (page 10, line 217-219): 

“The patient has undergone a hemicolectomy and specimen from the tumor of the cecum revealed BL with a reduced number of apoptotic bodies and starry sky appearance in HP.”

Comment 9: Figure 1: each square of the histological image should have its letter (1 a.b.c.d), while figure 1b and 1c should become figure 2 and 3. Doing so, line 230 clarifications are no longer necessary (“upper row” and so on). The EE image has a low quality. An inset with an evident blastoid morphology should be added. The CD38 image could benefit from pointers or arrows to focus on plasma cells, BL cells and T lymphocytes.

Response: We agree with this and we have prepared 3 separated figures (Figure 1, Figure 2 and Figure 3) from the former big one in the revised manuscript as requested. Figure 1: each square of the histological image has its letter (1 A.B.C.D). The morphological details with “blastoid morphology”of BL cells are shown in Figure 7, Case 2 (as suggested by the Reviewer#2). Therefore we have omitted the morphological details in Figure 1 as requested by the Reviewer #1 (Reviewer#1 suggestion “an inset with an evident blastoid morphology should be added”). Additionally, we have inserted arrows describing CD38 expression on plasma cells, on BL and on T lymphocytes. The description of a Figure 1B on CD38 expression is as follows (page 12, lines 250-252):

„The IHC test shows differences in CD38 staining between plasma cells (the strongest) (green arrows), BL cells (strong) (blue arrows), and T lymphocytes (the weakest, partially negative) (red arrows)”.

Comment 10: Figure 4b has a low quality. 

Response: We apologize if our original Figure 4b did not show clearly all signals. Unfortunately, the number of metaphases in Case 2 is low and they are of poor quality. However, we have made a revision of Figure 4b ( Fig 8B in the revised version of the manuscript). We have picked up a new metaphase of higher quality of MYC/IGH fusion signal. 

Comment 11: Lines 456-467: this part should be integrated into conclusions.

 Response: We appreciate the reviewer’s suggestion, but we would prefer to preserve lines 456-467 (pages 23-24, lines 521-533 in in the revised version of the manuscript) in Discussion section. We have created Conclusions section, because we had the intention to present the concentrated short summary, regardless of summary presented in Discussion section. To reconcile this, we have added “In summary” at the beginning of this part:

“In summary, to the best of our knowledge, this is the largest study devoted to cryptic MYC insertions in consecutive mainly adult suspBL patients, routinely diagnosed by HP/IHC and FNAB/FCM/CC/FISH examinations at a single institution. We confirmed that cryptic MYC insertions in BL are extremely rare but not incidental. In our large group of patients clinicopathologically suspected of BL, the frequency of this aberration was 1.9% and constituted 17% of MYC-negative suspBL. The remaining cases of MYC-negative suspBL were represented by BLL,11q. We detected the insertions through chromosome analyses and performed NGS examination of these alterations, which will extend our knowledge of the molecular features of very rare BL MYC insertions. The insertions we described were observed in sBL patients and resulted in cryptic MYC/IGH fusions. In one case, the breakpoint of the MYC was typical of IGH/MYC fusions in sBL, contrary to the other case in which the MYC break was as in variant IG/MYC fusions of sBL. Despite the rarity of MYC insertions, we believe that our study will substantially add to the understanding of MYC-negative BL and BLL,11q.

Comment 12: Supporting methods, immunohistochemistry: “...antigen-retrieval technique WERE...”

Response: Thank you for pointing this out. A fragment of the text in the Supporting information has been changed in the revised manuscript as requested:

“and, if necessary, antigen-retrieval technique were applied for each monoclonal antibody according to the manufacturer’s instructions”.

Comment 13: Supporting methods, flow cytometry: the sentence “Four to ten separate needle passes…” is potentially confusing. Maybe it’s better to say “Within the context of a single FNAB, four to ten separated needle passes…”.

Response: Thank you for pointing this out. A fragment of the text in the Supporting information of the revised manuscript has been changed according to Reviewer#1 suggestion as requested:

 “Within the context of a single FNAB, four to ten separated needle passes within a lymph node or tumor and three or four passes within abdominal mass provided adequate cellular material”.

Comment 14: S1 table: column1-line2 should be named “Total karyotype analysis”.

Response: Thank you for pointing this out. We have change the text according to Reviewer#1 suggestions. We have also sorted the abbreviations in the order of appearance:

Karyotype

 BL

(no of cases) BLL,11q

(no of cases) BL,MYCR/11q 

(no of cases)

Total karyotype analysis 

 73 10 5

t(8;14)(q24;q32)

 60 0 3

t(8;22)(q24;q11)

 7 0 2

t(2;8)(p11;q24)

 1 0 0

11q duplication/deletion 

 0 10 5

Normal karyotype/karyotype without t(8;V)

 5 0 0

Comments from Reviewer #2: 

MAJOR ISSUES

Comment 1: In both clinical cases (case 1 and case 2), the age of the patients, the ethnicity and the HIV status were reported while the EBV status lacks. It would be important to know if it has been determined or not, in order to be able to correlate the data obtained from the study with this data as well.

Response: Thank you for suggestion. We have supplemented the data on possible expression of EBV RNA in both cases in the revised manuscript as requested:

a) in Methods section (in Histopathology and immunohistochemistry, page 6, lines 121-124): 

“Latent membrane protein 1 (LMP1) expression by IHC and Epstein–Barr virus (EBV) small nuclear RNA transcripts (EBER) by in situ hybridization (ISH) method was performed in some patients as described previously [7].”

b) in Results section (Case 1, pages 10-11, lines 219-222): 

“IHC showed EBV-positive classic MYC-positive BL immunostaining for CD20+/CD10+/BCL6+/ BCL2−/MYC+ strong,100%/ LMO2−/MUM1−/CD38+strong/ EBER+/ EBV-LMP1−/CD43−/CD44−/CD56−/Ki-67 index > 98%/CD3−/CD5−/TdT− (Fig 1).”

c) in Results section (Case 2, page 15, lines 327-329): 

“The IHC showed EBV-negative classic MYC-positive BL (but partial BCL2±weaker positive) immunostaining for CD20+/CD10+/BCL6±/ MYC+strong,100%/ LMO2−/CD38+/ EBER−/EBV-LMP1−/MUM1−/CD43−/CD44−/ CD56−/Ki-67 index > 98%/CD3−/CD5−/ TdT− (Fig 7D).”

Due to the fact that both the discussed BL cases with MYC insertion differed in EBV status, this observation was not discussed in the discussion.

Comment 2: Unlike case 1, in case 2 clinical presentation and pathomorphological features are only described in the text. You should add some pictures (histological, cytological, radiological) as well.

Response: Thank you for suggestion. According to the Reviewer's request, we have transferred the flow cytometric figure of Case 2 from Supporting information to the Results section (Figure 6) (pages 15-16, lines 333-353) and we have supplemented Case 2 with histopathological pictures and morphological details (Figure 7) (“An inset with an evident blastoid morphology should be added” - detailed assessment of BL cytomorphology as suggested by the Reviewer#1)(page 16, line 354-365):

“Fig 6. Flow cytometry immunophenotyping including analysis of CD38 expression of Case 2. 

FCM analysis of BL cells from the peritoneal fluid. A: Forward scatter/side scatter dot plots present both small normal T lymphocytes (red cells) and larger lymphoma cells (green cells) with a reduce number of apoptotic bodies (marked by circles). B-D: BL expresses: CD20/CD19/CD22 (with median fluorescence intensity (MFI) of CD20> CD19> CD22), as shown by monoclonal antibodies conjugated with the same fluorochrome, APC-A as well as CD45+weaker/HLADR+. E-I: BL expresses a homogeneous phenotype of germinal center origin (CD81+higher/CD10+/CD38+higher/CD44– but BCL6 negative. H (enlarged dot plot): FCM-based analysis of MFI of CD38 expression in BL. MFI of CD38 expression on BL (873) in R1 was higher (CD38(+)higher) compared to normal T-lymphocytes (36) in R2 and apoptotic bodies (599) (in a circle). J-P: BL expresses CD54+higher/CD305+higher/ BCL2±weaker (very low expression on a small subpopulation of cells) but is negative for CD62L/kappa/lambda with a restricted expression of IgM+/IgD+ heavy immunoglobulin chain. In addition, CD71+++ expression is detected in 100% of BL cells. Antigen expression of few macrophages was marked with a pink asterisk (CD10/CD38/CD44/CD54/CD71/CD305). Antigen expression of BL cells is compared to the expression on a subpopulation of normal T-lymphocytes (most antigens) (i.e. CD38/CD43/CD44/CD45/CD54/CD81/BCL2) and on macrophages (i.e. CD54/ CD305) of the tumor and described as + higher for an antigen with a higher expression in BL cells compared to normal lymphocytes/ macrophages in 100% of cells; +, positive in 100% of BL cells; + weaker, for an antigen with a weaker expression than in lymphocytes/macrophages in 100% of cells; ± weaker, for an antigen with a weaker expression in BL cells compared to normal lymphocytes/macrophages in >20% to <100% of BL cells; –, no expression (i.e. expression in <20% BL cells). Dot-plots.”

 “Fig 7. Pathomorphological features of Case 2.

A: A monomorphic population of BL cells in the absence of apoptotic bodies in the background is visible in the cytological smear obtained from the FNAB of the liver tumor. Cytologic features with relatively uniform round nuclei, more cells with single, central nucleoli, and thin rims of cytoplasm – “small immunoblast” (cytological smear stained with HE, original magnification, 800×). B: A trephine biopsy showing heavy infiltration with BL. C: Gastric tissue biopsy showing heavy infiltration with BL. B-C: Both these images revealed BL with the reduced number of apoptotic bodies and starry sky appearance in HP. High magnification showing a “squaring off” of the cytoplasm. Also note the slight nuclear irregularity and more cells with single, central nucleoli (B-C: paraffin section stained with HE, original magnification, 800×). D: MYC protein immunostaining is strongly expressed by all BL cells. C-D: The images show stomach wall glands, which are also MYC positive (D) (D: original magnification 800×).”

We have shown a second flow cytometry Figure 6 using a broad panel of antibodies to explain our an innovative original flow cytometry-based diagnostic algorithm, enabling BL and BLL,11q diagnosis within 1.5 hours following fine needle aspiration biopsy.

Comment 3: Figure 4B: it is very difficult to distinguish the FISH signals reported in the description. You should provide an image with better definition.

Response: We apologize if our original Figure 4B did not show clearly all signals. Unfortunately, the number of metaphases in Case 2 is low and they are of poor quality. However, we have made a revision of Figure 4B (Fig 8B in the revised version of the manuscript). We have picked up a new metaphase of higher quality of MYC/IGH fusion signal.

MINOR ISSUES

Comment 1: Page 4, lines 85-88: you should report the part concerning the re-definition of “BLL, 11q” cases diagnosed before the WHO review in 2016 in the results section.

Response: We appreciate the reviewer’s suggestion, however, we think that the part concerning redefinition of “BLL,11q” should stay in Materials and methods section. We have added the following information about the treatment (page 4, lines 90-92): 

“All BLL,11q cases diagnosed before the latest revision of the 2016 WHO classification were primarily diagnosed and treated as MYC-negative BL at our Institute”.

Comment 2: Page 5, line 99: you should define the exact number of cases in which FISH was performed on cultured cells and the exact number of cases in which FISH was performed on formalin-fixed paraffin-embedded (FFPE) sections.

Response: Thank you for this suggestion. We have added this information in Materials and methods section (page 5, lines 104-105 in the revised version of the manuscript):

“FISH analysis was performed on cultured cells in 104/108 patients. In 4/108 patients, a formalin-fixed paraffin- embedded (FFPE) tumors were used. In six patients both type of samples were used.”

 Comment 3: Page 5, line 117: you should modify the title of the paragraph, also including the fact that a morphological evaluation on cytological smears was performed.

Response: Thank you for this suggestion. As suggested by the Reviewer#2, the title of the paragraph was extended to include morphological assessment of lymphomas (page 6, line 125):

“Flow cytometry with cytological smears evaluation”

Comment 4: Page 7, lines 151-156: in “Materials and methods” it is reported that PCR and Sanger sequencing reactions were performed to verify the existence of the two interchromosomal translocations detected by NGS. You should report the results obtained in the results section.

Response: Thank you for this suggestion. We have added the relevant information regarding PCR and Sanger sequencing in the Results section and in the figure legends in the revised version of the manuscript :

a) page 13, line 277-280: “The interchromosomal translocation analysis with our original CTX-explorer software (see Material and methods for details), followed by PCR and Sanger sequencing, showed that the breakpoint on chromosome 8 was located 158 kb downstream of 3′MYC, in the PVT1 region (chr8:127,901,209) (Figs 4 and 5).”

b) page 14, line 296-298: “D: Detailed breakpoints identified by PCR and Sanger sequencing: the break on chromosome 8 maps to the PVT1 region; the break on chromosome 14 is located 1.6 kb upstream of the Sμ switch region.”

c) page 17, lines 371-374: “The usage of the CTX-explorer app for identifying chromosomal breakpoints, followed by PCR and Sanger sequencing, revealed that the breakpoint on chromosome 8 was located 43 kb upstream of the 5′MYC (chr8:127,692,550), and the breakpoint on chromosome 14 was in a diversity IGH region, 2 kb downstream of 3′IGHD2-2 (chr14:105,914,873) (Figs 5 and 8 and S1B Fig).”

d) page 17, lines 380-382: “C: Detailed breakpoints identified by PCR and Sanger sequencing: the break on chromosome 8 maps 43 kb upstream of the 5′MYC; the break on chromosome 14 is 2 kb downstream of 3′IGHD2-2.”

Comment 5: Page 7, line 161: you should express in brackets the exact number of cases in which FISH was performed and the exact number of cases in which classical cytogenetic was performed.

Response: We agree and we have added this information in Results section (page 8, lines 173-174 in the revised version of the manuscript):

“Both the CC and FISH were made in 86/108 patients. In the remaining 22/108 patients, FISH (20/108) or CC (2/108) were carried out.”

Comment 6: Page 10, line 198: you should also express the extended version of the “CRP” acronym.

Response: Thank you for pointing this out. The acronym "CRP" (C-reactive protein) has been extended in Results section (page 10, line 214 in the revised manuscript) as requested:

“His serum lactate dehydrogenase (LDH) (940 IU/L, n < 240), β2-microglobulin (4.44 ng/L, n = 0.7–1.8), d-dimer (1247 ng/mL, n < 500), C-reactive protein (CRP) (36.2 mg/L, n < 5 mg/L) and fibrinogen (3.59 g/L, n = 1.7–3.5) levels were elevated, with the biochemical features of renal failure, an ECOG performance status of 0, and Ann Arbor stage of IVA without B symptoms.”

Comment 7: Page 14, lines 294-295: immunohistochemical evaluation of BCL6 is reported twice. Regarding BCL6 results reported in the text, you should also check the agreement with the same data reported in the diagrams of figure S2B.

Response: Thank for pointing this out. The error of repeated description of BCL6 by flow cytometry (not IHC) in the text (Results section) has been fixed in the revised manuscript as requested (page 14, lines 312-317):

“BL cells from the peritoneal fluid and liver tumor were positive for CD45weaker/CD20bright/CD19bright/CD22 (with an order according to MFI of CD20 > CD19 > CD22)/CD10/CD38higher (with an MFI of 873 for CD38, compared to an MFI of 36 on T lymphocytes /CD81higher/BCL6/CD79β/HLA-DR/CD43weaker/ CD49dweaker/CD52higher/CD54higher/CD305/ MYC and surface immunoglobulin (IgD/IgM), while they were negative for CD5/CD8/CD11c/CD23/CD25/CD44/CD16&CD56/CD56/CD62L/CD200/ IgG/λ/κ.“

Lack of BCL6 expression on BL cells by flow cytometry in Case 2 corresponds to Figure 6G, where BL cells are BCL6 negative.

Comment 8: Pages 19-20, lines 411-428: you should add that this analysis was performed in the “Materials and methods” section too.

Response: Thank you for this suggestion. The relevant information has been added to “Materials and methods” section: page 7, lines 158-161:

“In order to verify the sensitivity and specificity of the CTX-explorer-based breakpoint predictions, our results obtained with this piece of software were compared with the output of Breakdancer [25] and Delly [26], open-source programs developed by other research teams.”

We would like to thank the referee again for taking the time to review our manuscript. 

Sincerely,

Renata Woroniecka

02-12-2021

---

## [Decision Letter · Decision Letter 1]

7 Jan 2022

PONE-D-21-27800R1Cryptic *MYC* insertions in Burkitt Lymphoma: new data and a review of the literaturePLOS ONE

Dear Dr. WORONIECKA,

Thank you for submitting your manuscript to PLOS ONE. After careful consideration, we feel that it has merit but does not fully meet PLOS ONE’s publication criteria as it currently stands. Therefore, we invite you to submit a revised version of the manuscript that addresses the points raised during the review process.

We look forward to receiving your revised manuscript.

Kind regards,

Vincenzo L'Imperio

Academic Editor

PLOS ONE

Journal Requirements:

**Comments to the Author**

Reviewer #1: The authors have assessed the issues clarifying doubts and appropriately modifying the paper, therefore in my opinion the article can be submitted for publication.

Line 173: typing error (condacted).

Reviewer #2: I would like to thank the authors for all the answers to my comments and for the revision of the manuscript. In my opinion there are still some issues that need to be addressed before publication:

-Page 4, lines 84-87: this part about epidemiological data of the population should be reported in the results section.

-Figure 6: the diagrams are of low quality, small and difficult to read.

-Page 15, line 326; page 16, line 354 and figure 6: you should check the agreement between BCL6 flow cytometry results in the text and in figure 6 description and diagrams.

→ Page 15, line 326: “BL cells from the peritoneal fluid and liver tumor were positive for…BCL6”

→ Page 16, line 354 and figure 6: “E-I: BL expresses a homogeneous phenotype of germinal center origin (CD81+higher/CD10+/CD38+higher/CD44–) but BCL6 negative”.

---

## [Author Response · Author response to Decision Letter 1]

25 Jan 2022

Dear Prof. Vincenzo L’Imperio,

We appreciate you and the reviewers for your precious time in reviewing our paper and

providing valuable comments once again. All comments were helpful for revising and improving our paper and we have taken them fully into account in revision.

We are submitting the corrected manuscript with the suggestions incorporated to the text.

Our responses to all the comments are as follows: 

(All page and line numbers refer to the manuscript file without track changes - file name “Manuscript”)

Comments from Reviewer #2: 

Comment 1: Page 4, lines 84-87: this part about epidemiological data of the population should be reported in the results section.

Response: Thank you for suggestion. According to the Reviewer's request, we have transferred the part about epidemiological data from Materials and methods to the Results section (pages 8, lines 169-172 in the revised version of manuscript):

 “This group of patients consisted of 102 adults with median age of 35 years (range, 19-79 years) and 6 children with median age of 8 years (range, 3-12 years). Among adult patients, 81 were male and 21 were female (ratio, 3.86:1). Among pediatric patients, 5 were male and 1 was female (ratio, 5:1).”

Comment 2: Figure 6: the diagrams are of low quality, small and difficult to read.

Response: According to the Reviewer's #2 suggestion, the diagrams (immunophenotype descriptions) in Figure 6, which were of low quality, small and difficult to read, have been changed in the revised manuscript as requested. We decided to change the colour (from blue to black), and to increase the font size to get a higher quality figure. In a similar way, we changed the font of Figure 3.

Comment 3: Page 15, line 326; page 16, line 354 and figure 6: you should check the agreement between BCL6 flow cytometry results in the text and in figure 6 description and diagrams.

Comment 4: Page 15, line 326: “BL cells from the peritoneal fluid and liver tumor were positive for…BCL6”

Comment 5: Page 16, line 354 and figure 6: “E-I: BL expresses a homogeneous phenotype of germinal center origin (CD81+higher/CD10+/CD38+higher/CD44–) but BCL6 negative”.

Response to comments 3, 4, 5: We sincerely apologize for the error in the text describing the BL immunophenotype (Page 15, line 326: “BL cells from the peritoneal fluid and liver tumor were positive for…BCL6”) in terms of BCL6 expression in the flow cytometric test. A fragment of the text concerning BCL6 flow cytometry results has been changed according to Reviewer#2 suggestion (pages 14-15, lines 312-318 in the revised manuscript):

“BL cells from the peritoneal fluid and liver tumor were positive for CD45weaker/CD20bright/CD19bright/CD22 (with an order according to MFI of CD20 > CD19 > CD22)/ CD10/CD38higher (with an MFI of 873 for CD38, compared to an MFI of 36 on T lymphocytes) /CD81higher/CD79β/HLA-DR/CD43weaker/CD49dweaker/CD52higher/CD54higher/CD305/MYC and surface immunoglobulin (IgD/IgM), while they were negative for CD5/CD8/CD11c/CD23/CD25/CD44/ CD16&CD56/CD56/CD62L/CD200/IgG/λ/κ and BCL6.”

We would like to thank the referee again for taking the time to review our manuscript. 

Sincerely,

Renata Woroniecka

25-01-2022

---

## [Editor Report · Decision Letter 2]

2 Feb 2022

Cryptic *MYC* insertions in Burkitt Lymphoma: new data and a review of the literature

PONE-D-21-27800R2

Dear Dr. WORONIECKA,

We’re pleased to inform you that your manuscript has been judged scientifically suitable for publication and will be formally accepted for publication once it meets all outstanding technical requirements.

Kind regards,

Vincenzo L'Imperio

Academic Editor

PLOS ONE

---

## [Editor Report · Acceptance letter]

4 Feb 2022

PONE-D-21-27800R2 

Cryptic *MYC* insertions in Burkitt Lymphoma: new data and a review of the literature 

Dear Dr. WORONIECKA:

I'm pleased to inform you that your manuscript has been deemed suitable for publication in PLOS ONE. Congratulations! Your manuscript is now with our production department. 

Kind regards, 

on behalf of

Dr. Vincenzo L'Imperio 

Academic Editor

PLOS ONE